

# Seasonal forecasting of hydrological drought in the Limpopo basin: A comparison of statistical methods.

Mathias Seibert[1], Bruno Merz[1,2], and Heiko Apel[1]

[1]GFZ - German Centre for Geosciences, Telegrafenberg, 14473 Potsdam, Germany
[2]University of Potsdam, Potsdam, Germany

*Correspondence to:* Mathias Seibert (mathias.seibert@gmail.com)

**Abstract.** The Limpopo basin in southern Africa is prone to droughts, which affect the livelihoods of millions of people in South Africa, Botswana, Zimbabwe, and Mozambique. Seasonal drought early warning is thus vital for the whole region. In this study, the predictability of hydrological droughts during the main runoff period from December to May is assessed with statistical approaches. Three methods (Multiple Linear Models, Artifical Neural Networks, Random Forest Regression Trees) are compared in terms of their ability to forecast streamflow with up to 12 months lead time. The following four main findings result from the study. 1) There are stations in the basin at which standardised streamflow is predictable with lead times up to 12 months. The results show high interstation differences of forecast skill but reach a coefficient of determination as high as 0.73 (cross validated). 2) A large range of potential predictors is considered in this study, comprising well established climate indices, customised teleconnection indices derived from sea surface temperatures, and antecedent streamflow as proxy of catchment conditions. El-Niño and customised indices, representing sea surface temperature in the Atlantic and Indian Ocean, prove to be important teleconnection predictors for the region. Antecedent streamflow is a strong predictor in small catchments (with median 42% explained variance), whereas teleconnections exert a stronger influence in large catchments. 3) Multiple linear models show the best forecast skill in this study and the greatest robustness compared to artificial neural networks and Random Forest regression trees, despite their capabilities to represent non-linear relationships. 4) Employed in early warning the models can be used to forecast a specific drought level. Even if the coefficient of determination is low, the forecast models have a skill better than a climatological forecast, which is shown by analysis of receiver operating characteristics (ROC). Seasonal statistical forecasts in the Limpopo show promising results, and thus it is recommended to employ them complementary to existing forecasts in order to strengthen preparedness for droughts.

## 1   Introduction

Drought is a slowly progressing phenomenon which is challenging to detect ahead. As a result, drought management frequently remains crisis management, which is limited to fighting drought when impacts have already started to unfold. A more desirable reaction is the conversion of crisis management to risk management (Wilhite and Hayes, 2000). This is a challenging process in which drought forecasting with long lead time is recommended in order to adopt mitigation actions and raise preparedness (Vicente-Serrano et al., 2012a). Forecasting products should be tailored to the end users' needs, such as water resources





managers (Winsemius et al., 2014; Masih et al., 2014). The forecast information must satisfy the need for a thorough drought assessment without overwhelming end users with high complexity. Seasonal forecasts have a high uncertainty, therefore it is an important task to convey the skill of the forecast system, and for end users to include uncertainty information in the decision making process. This can be achieved, for example, by providing probabilistic drought forecast information.

The Limpopo basin in southern Africa (ca. 408000 km²) is strongly affected by droughts, which had severe impacts on agriculture, economy and food security in southern Africa, for example, in the early 1980s and 1990s (Love et al., 2010; Rouault and Richard, 2003; Rouault, 2005; Masih et al., 2014; FAO, 2004). The Limpopo basin is a highly modified catchment, where irrigation demands by agriculture are high and might even exceed supply in parts of the basin (FAO, 1997). Southern African water resources are regarded highly affected by seasonal variability, a fact that is likely to be exacerbated by climate

change (Kusangaya et al., 2014). Zhu and Ringler (2010) estimated a decrease in Limpopo streamflow by 2030 due to climate change, which is contradictory to studies who found increases of precipitation (Tadross et al., 2005) and runoff (Li et al., 2015) in parts of southern Africa, including vast parts of the Limpopo basin. The common climate change paradigm "dry gets drier and wet gets wetter" only seems to hold for about 10.8% of the Earth's terrestrial surface according to Greve et al. (2014), excluding the Limpopo region. Hence, it seems that the climate change impact in the Limpopo basin remains very uncertain

and might exhibit a stronger effect on precipitation variability than on average precipitation (Tadross et al., 2005). The high inter-annual variability of precipitation and the tense condition of water resources require improvements in water management in the riparian states (South Africa, Botswana, Zimbabwe and Mozambique) who cooperate within the "Limpopo Watercourse Commission" since 2003. This study analyses the annual to seasonal predictability of (seasonal) hydrological drought in the Limpopo basin using statistical methods which could improve the preparedness and help mitigate drought disasters.

In order to understand hydrological droughts and to cope with them properly, an appropriate drought indicator has to be selected and forecasted (Wetterhall et al., 2015; Winsemius et al., 2014). In the Limpopo basin, the main rainy season runoff lasts from December to May and the total streamflow of the period is an adequate indicator for hydrological droughts. In this study the standardised streamflow index (SSI) is used as hydrological drought index (Vicente-Serrano et al., 2012b). Standardisation of streamflow is less common than for precipitation (Mishra and Singh, 2010), but nevertheless useful for

two reasons: First, it facilitates the comparison of droughts at different stations. Second, the standardised indicator is normally distributed and has, therefore, a higher sensitivity to droughts compared to original streamflow which is often strongly positively skewed.

    Statistical streamflow forecasting is challenging due to the complexity of the signal and the underlying processes, especially in highly modified catchments such as the Limpopo basin (FAO, 2004). The streamflow signal integrates meteorological,

hydrological and anthropogenic effects, such as irrigation and water storage, thus interlacing hydrological drought and water scarcity (Van Loon and Van Lanen, 2013). Anthropogenic effects (e.g. operation of dams and irrigation) are typically time-varying and can be considered in hydrological models (Trambauer et al., 2014). Thereby, it is possible to separate drought from water scarcity (Van Loon and Van Lanen, 2013) by simulating naturalised streamflow. In a statistical approach as presented here, the anthropogenic effect is not accounted for and therefore increases prediction uncertainties.





The processes in the atmospheric circulation have a chaotic component, that is not susceptible to prediction, but predictability can be deduced from the land-atmosphere and land-ocean interactions. The latter can be represented by teleconnections to sea surface temperatures (SST), which is a common approach in both tropical and humid climates. In more dry climate zones the land-atmosphere interaction, and therefore the land surface moisture condition, is likely to be more important (Koster et al., 2000), since atmospheric moisture is recycled over the land surface (Gimeno et al., 2010). It can be expected that both SST and land surface conditions are important factors in the Limpopo basin, because it reaches from the ocean to very arid regions in Botswana.

The term teleconnection refers to the influence of sometimes remote Ocean regions on atmospheric variables, such as moisture content or precipitation. Past studies on southern African precipitation found predictability based on El-Nino, the Indian and the Atlantic Ocean (Reason et al., 2006; Landman et al., 2005; Landman and Mason, 1999). However, the atmospheric circulation is very complex, sometimes having the effect that even strong El-Niño events do not propagate to the region (Thomson et al., 2003). A reason might be that the ocean region south of Africa is the major source for precipitation in southern Africa (Gimeno et al., 2010). This region is characterised by a chaotic collision system of the warm Agulhas and the cold Antarctic circumpolar ocean current (see Figure 1) (Peterson and Stramma, 1991). In the collision process warm Agulhas eddies can form, maintaining higher evaporation until they dissipate. There are more complex effects such as the Darwin sea level pressure (Manatsa et al., 2007), the linkage of ENSO with the Indian Ocean Dipole (Yuan and Li, 2008) or the stratospheric quasi-biennial oscillation (Jury, 1996) and even the Antarctic Ozone depletion (Manatsa et al., 2013). Despite that complexity, SST teleconnections remain the preferred choice of predictors in seasonal forecasting (Landman et al., 2005; Landman and Mason, 1999; Funk et al., 2014). In this study widely used climate indices are complemented with customised indices resulting from a composite and correlation analysis of SSTs in the Indian and Atlantic Ocean.

Many methods have been applied in drought forecasting (Mishra and Singh, 2011). Three models are chosen for comparison in this study. First, multiple linear models (MLM) which are widely used in similar studies (Diro et al., 2011, e.g.). They are however limited to linear combinations of predictors. Artificial Neural Networks (ANN) are applied as a second method. They are flexible nonlinear models and have been applied successfully in several seasonal prediction studies (Mwale et al., 2004; Morid et al., 2007; Mishra and Desai, 2006). In addition, we develop Random Forest Regression Tree models (RFOR) (Breiman, 2001). They are particularly suited for representing conditional relationships in complex data including non-linearities. Random Forest regression trees have only rarely been applied for seasonal drought forecasting (Chen et al., 2012). These data-driven approaches are useful for seasonal forecasting in regions where hydrological observations are available, but additional data characterising the catchments is limited.

A recent publication by Trambauer et al. (2015) presented forecasting results for the Limpopo basin achieved by a chain of process-based models, namely the hydrological model PCR-GLOBWB (van Beek, L. P. H. and Bierkens, 2009) with input from the seasonal forecasting system S4 (Molteni et al., 2011) and Reanalysis data ERA-Interim (Dee et al., 2011) by ECMWF. The forecasting system for DJFMAM streamflow exceeded climatological forecasts (climatology) with "moderate skill for all lead times" up to 5 months (forecast in December) (Trambauer et al., 2015). To parameterise such models is challenging in data sparse regions such as southern Africa (Trambauer et al., 2014). Compared to forecasts based on simulation models, statistical





forecast models require less input data and computational power. The main requirement for model development is a sufficiently long record of relevant drought indicators. In summary, both approaches have their advantages and disadvantages. Here, we evaluate the predictability of hydrological drought in a data-driven approach, which can serve as a baseline for other seasonal forecast systems. Special care is taken of the predictor selection, model validation and forecast verification process for the use

of the forecast models in a drought early warning system. We present the forecasting skill for hydrological drought during the main rainy season runoff from December to May achieved with the three selected statistical models.

## 2 Data and Methods

### 2.1 Study area: Limpopo basin

The Limpopo basin is located in southern Africa with the riparian states South Africa, Botswana, Zimbabwe and Mozambique,

where it flows into the Indian Ocean and covers an area of approx. 400000 km² (Figure 1). The climate is dominated by hot steppe climate, while the southernmost regions reach into the warm temperate climate zone of South Africa and the eastern region comprises parts of the savanna climate in Mozambique. The highest mountains of the Waterberg mountain range in South Africa reach ca. 2300 m in elevation. The rainy season usually lasts from October to March. The average annual rainfall ranges from ca. 250 to 1050 mm with 530 mm in average, but with high interannual variation, which makes drought a common

natural hazard. Mean annual runoff is approx. 4550 million $m^3$ $a^{-1}$(station Chókwè). Rainfed farming and grazing is very common, but commercial irrigation farming is also widespread, so that irrigation is the most important water usage with about 50% of total water use (FAO, 2004). Intrabasin and interbasin water transfers exist in South Africa (interbasin transfers from Incomati, Usutu and Orange rivers) and Botswana (intrabasin transfers). Water use and storage heavily affect streamflow with the effect that for example in the Matlabas subcatchment only approx. 5% of the naturalised mean annual runoff are recorded

(FAO, 2004, Table 8). The total dam capacity is ca. 2500 million m³ and the dam capacity per subcatchment often exceeds mean annual streamflow (Table 1). Hence, many of the streamflow time series are heavily affected by water use and management.

### 2.2 Data

The data base of this study comprises streamflow data, which serves as both predictand and predictor, climate indices and gridded sea surface temperature anomalies (potential predictors). The Global Runoff Data Centre (GRDC, 2011) provides

streamflow from all countries in the Limpopo basin. This data is extended by the runoff observations available from The Department of Water Affairs of the Republic of South Africa (DWAF) and Mozambique Regional Administration of Waters in the South (ARA-Sul). A subset of 16 stations (Figure 1 and Table 1) satisfies the following conditions:

- *Record length* of at least 30 years (360 observations at monthly resolution),

- *Completeness* of at least 90% in the observation period.



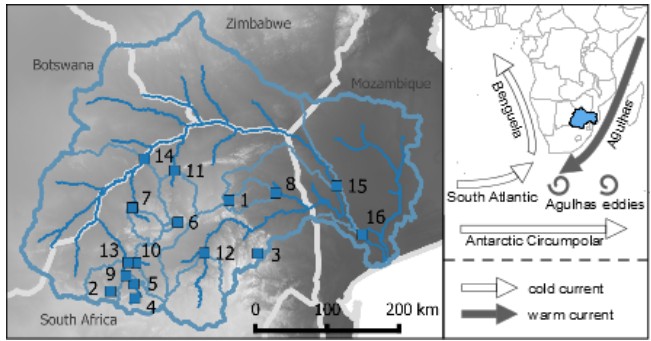

**Figure 1.** Location of Limpopo basin and streamflow stations. Streamflow stations with subbasins numbered according to table1 and elevation (max. 2300 m) as background (left). Location of the Limpopo basin within Sub-Saharan Africa and schematic of ocean currents (right).

The stations selected for this study are mainly located in the South African part of the basin where data availability is better and the conditions are met (Figure 1). The HydroSHEDS dataset (Lehner et al., 2008) is used to derive catchment outlines per station and catchment areas. Dam capacities are collated from the DWAF database (http://www.dwaf.gov.za/Hydrology/).

Several prominent atmospheric indices are acquired from the online resources provided by the Climate Prediction Center of
the National Oceanic and Atmospheric Administration (NOAA). To represent the influence of the El Niño Southern Oscillation (ENSO), the SST indices of regions Niño-1+2, Niño-3, Niño-4, Niño-3.4 are compiled in addition to the Trans Niño and the Oceanic Niño index (see Table 2 for a detailed list). These indices form the basis of the potential predictors. This set of widely used indices is augmented with customised predictors based on analysing the sea surface temperature data set HadISST 1.1 (Rayner, 2003) provided by the British Metoffice.

**2.3  Hydrological drought predictant: Standardised streamflow index**

The use of standardised streamflow translates into a drought metric independent of catchment size, climatology and streamflow characteristics (Lorenzo-Lacruz et al., 2012). Furthermore, the strength of events anomalies can easily be compared between very different catchments. For the purpose of seasonal drought forecasting, the interannual variability of low flow is more important than in a general streamflow forecast. The distribution of streamflow is usually right-skewed, hence high extremes
can have a large effect in the model training process. Standardisation transforms the original flow distribution into a normal distribution with zero mean and standard deviation of one. Thus, it is likely that the models are more sensitive to low flow variability, when trained with standardised streamflow. However, only a few hydrological studies use standardised streamflow, e.g. Modarres (2006). In meteorological studies, forecasting of standardised precipitation is more frequent, e.g. Mishra and Desai (2005); Morid et al. (2007); Belayneh et al. (2014). Another beneficial aspect of forecasting standardised indices is that
the transformed variables are normally distributed and defined for $\mathbb{R}$, whereas precipitation and streamflow are defined for $\mathbb{R}^{\geq 0}$, only. Thus, corrections, normally applied to prevent undefined forecasts, such as precipitation below zero, are not necessary.





**Table 1.** Streamflow stations included in the analysis: Observation period, average annual streamflow volume ($Q_{ann}$) in [M m$^3$ a$^{-1}$], dam capacity ($V_{Dam}$) in [M m$^3$], dam capacity relative to mean annual flow volume ($V_{Drel}$), catchment area (Area) in [km²]. Dam capacities are estimations based on available information.

| | Station | Time period | | | $Q_{ann}$ | $V_{Dam}$ | $V_{Drel}$ | Area |
|---|---|---|---|---|---|---|---|---|
| 1 | Woodbush | Jun 1977 | - | Feb 2012 (34.2 yrs) | 8.53 | 1.94 | 0.23 | 12 |
| 2 | Rietvallei | Jan 1971 | - | Mar 2012 (39.6 yrs) | 2.90 | 0.00 | 0.00 | 15 |
| 3 | Naauwpoort | Mar 1957 | - | Oct 2010 (47.1 yrs) | 10.57 | 14.22 | 1.35 | 87 |
| 4 | Hartbeeshoek | Oct 1964 | - | Mar 2012 (44.8 yrs) | 4.52 | 0.00 | 0.00 | 101 |
| 5 | Krokodilriver | Mar 1972 | - | Mar 2012 (40.1 yrs) | 185.46 | 0.00 | 0.00 | 215 |
| 6 | Doorndraai | Sep 1954 | - | Feb 2012 (57.4 yrs) | 8.24 | 44.20 | 5.36 | 409 |
| 7 | Mokolo | Aug 1980 | - | Feb 2012 (31.4 yrs) | 119.50 | 145.92 | 1.22 | 4315 |
| 8 | Letaba Ranch | Oct 1959 | - | Mar 2012 (45.4 yrs) | 100.77 | 235.60 | 2.34 | 4724 |
| 9 | Beestkraal | Mar 1951 | - | Mar 2012 (59.9 yrs) | 153.67 | 268.79 | 1.75 | 6032 |
| 10 | Klipvoor | Apr 1970 | - | Mar 2012 (42 yrs) | 118.70 | 134.16 | 1.13 | 6159 |
| 11 | Glen Alpine | May 1970 | - | Feb 2012 (41.8 yrs) | 101.40 | 67.47 | 0.67 | 11246 |
| 12 | Loskop Noord | Sep 1938 | - | May 2011 (44.7 yrs) | 230.14 | 960.68 | 4.17 | 16542 |
| 13 | Buffelspoort | Sep 1955 | - | Mar 2012 (56.5 yrs) | 131.27 | 487.45 | 3.71 | 20383 |
| 14 | Botswana | Apr 1971 | - | Feb 2012 (36.8 yrs) | 475.09 | 945.19 | 1.99 | 100977 |
| 15 | Combomume | Mar 1966 | - | Aug 2011 (41.1 yrs) | 3084.29 | 1311.16 | 0.43 | 259214 |
| 16 | Chókwè | Jul 1951 | - | May 2011 (56.8 yrs) | 4552.25 | 3252.13 | 0.71 | 343225 |

**Table 2.** Climate indices used as potential predictors

| Variable / Data set | Start | Source |
|---|---|---|
| Southern Oscillation Index (SOI) | 01.1951 | Climate Prediction Center of NOAA |
| Darwin sea level pressure | 01.1951 | Climate Prediction Center of NOAA |
| Tahiti sea level pressure | 01.1951 | Climate Prediction Center of NOAA |
| ENSO indices (ERSST) | 01.1950 | Climate Prediction Center of NOAA |
| ENSO indices (OISST) | 01.1982 | Climate Prediction Center of NOAA |
| North Atlantic Oscillation (NAO) | 01.1950 | Climate Prediction Center of NOAA |
| Indian Ocean Dipole Mode Index (DMI) | 11.1981 | Based on OISST Ver.2 (Reynolds et al., 2007) |
| Oceanic Nino Index (ONI) | 02.1950 | Based on ERSST.v3b (Smith et al., 2008) |
| Trans Nino Index (TNI) | 03.1870 | HadSST1.1 and OISST Ver.2 |
| NINO3.4 (HadSST) | 01.1871 | Climate Prediction Center of NOAA |




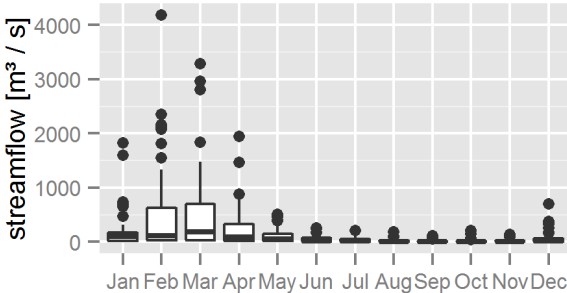

**Figure 2.** Boxplot of monthly streamflow at station Chòckwé.

Streamflow standardisation (Shukla and Wood, 2008; Vicente-Serrano et al., 2012b) is conducted using the algorithms implemented in the R-package SPEI version 1.6. This algorithm uses the Gamma distribution and unbiased Probability Weighted Moments as fitting method. The standardised streamflow indices (SSI) are calculated for each station at the scale of 6 months. $SSI_6^{May}$ of May at that scale covers the desired main runoff period from December to May, henceforth named $SSI_{DJFMAM}$

(Figure 2). The streamflow conditions are classified as drought, when SSI < -0.5, which has a 30.9% probability (given the normal distribution of SSI), thus is an approximation of the lower tercile extremes.

### 2.4 Potential predictors: Customised climate indices based on SSTs

Besides widely used climate indices, specific predictors are derived based on an analysis of past droughts and streamflow variability. SST fields and SSI are compared to detect ocean regions with predictive potential for streamflow in the Limpopo

basin. The analysis is limited to streamflow of the station Chockwe which has the largest catchment area of all stations. The ocean region is restricted to an area extending from latitudes 50° South to 25° North and longitudes 65° West to 115° East.

Composite analysis is used to identify ocean regions with predictive potential for droughts. Composites are generated for SST anomalies preceding drought during December to May defined by the drought threshold of $SSI_{DJFMAM} < -0.5$. Hence composites are calculated for every month from the November before DJFMAM to the previous year's December. Composite

maps are constructed by calculating the average SST field for the selected years. The resulting map shows the SST anomalies associated to droughts in the Limpopo basin and the respective significance levels tested with the Mann-Whitney test for two samples.

Additionally, correlation analysis is conducted with $SSI_{DJFMAM}$ and the SST field to identify ocean regions with predictive potential for streamflow variability. The significance of the Pearson correlations is calculated with $t = r\sqrt{\frac{N-2}{1-r^2}}$ using the

Student's t distribution with degrees of freedom $df = N - 2$, sample size $N$ and observed correlation $r$ by testing the null hypothesis: $\rho = 0$ (correlation of the general population).

Ocean regions that show correlations and composite anomalies with a significance level of 0.05 are chosen for the construction of potential predictors. Then, the region outlines are manually specified and defined rather generously as to cover the



**Table 3. Lead** time definition of $SSI_{DJFMAM}$ forecast: Time of **input** parameters (months abbreviated) and warning **issue** time.

| lead | 12 | 9 | 6 | 3 | 2 | 1 | 0 | -1 |
|---|---|---|---|---|---|---|---|---|
| input | prev. Nov | Feb | May | Aug | Sep | Oct | Nov | Dec |
| issue | prev. Dec | Mar | Jun | Sep | Oct | Nov | Dec | Jan |

anomaly regions resulting from different analyses. Then, indices are calculated by spatially aggregating the SST data to obtain time series with monthly means. Every index is calculated at three aggregation levels: 1, 3 and 6 months. A longer aggregation period indicates longer lasting anomalies of SST, while the one-monthly anomaly might capture short term effects.

## 2.5 Forecast model setup

The objective for the modelling is the predictability analysis of standardised streamflow using teleconnections and catchment conditions as predictors in data-driven approaches. Suitable statistical methods are compared by assessing the prediction performance and robustness for drought early warning with a leave-one-out cross-validation scheme. The adopted statistical methods are Multiple Linear Models (MLM), Artificial Neural Networks (ANN) coupled to the Genetic Algorithm (ANN-GA) and Random Forest Regression Trees (RFOR). MLMs are very common in modelling systems with linear relationships between

predictors and predictands (see 2.5.1). ANN-GA and RFOR are established data mining methods. Both have the advantage of allowing non-linear relationships. ANN are applied in this work in order to evaluate if the forecast quality of the MLM predictor combinations can be improved by allowing non-linear relations. In a similar study, where Australian rainfall was forecasted, Mekanik et al. (2013) achieved even better generalisation properties with ANN compared to MLM. ANN-GA and RFOR differ, among other aspects, in the type of results which are deterministic for ANN-GA and probabilistic for RFOR (see

details in 2.5.2 and 2.5.3 ). An overview of data flow, model validation and forecast verification is presented in Figure 3 and details are discussed in section 2.6.

The models are set up to predict $SSI_{DJFMAM}$ at the lead times of 1, 2, 3, 6, 9 and 12 months. We apply a strict definition of lead time as the difference (in months) between the availability of the forecast and the start of the predicted period. The resulting dates of forecast issue are presented in table 3. Some time is lost due to the dependency on external predictor data

sources, which are not available immediately, due to collation and processing operations. Thus, the forecast based on month $m$ would be available in the following month $m + 1$.

### 2.5.1 Multiple linear models

The first type of the data-driven models is the statistical multiple linear model (MLM). In MLM the dependent variable $y$ is related to linear combinations of the intercept $\beta_0$ , the predictors $x_1$ to $x_p$ with slope factors $\beta_1$ to $\beta_p$ and the error term $\varepsilon$:

$$y = \beta_0 + \beta_1 x_1 + \beta_2 x_2 + \cdots + \beta_p x_p + \varepsilon \qquad (1)$$




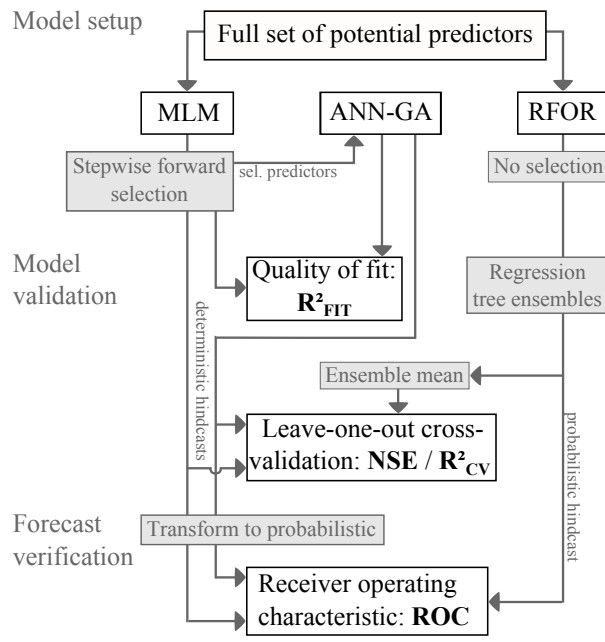

**Figure 3.** Modelling and validation scheme to account for the differences in models: Multiple linear models (MLM), artificial neural networks (ANN-GA) and Random Forest regression tree models (RFOR). The predictors selected for the MLMs are also used in the ANN-GAs, whereas RFOR works with the complete predictor set. MLM and ANN-GA are deterministic and require transformation to probabilistic form, whereas RFOR is probabilistic and is transformed to a deterministic value for the purpose of forecast comparison. Deterministic forecast skill is assessed using the coefficient of determination $R^2_{CV}$, the probabilistic properties are analysed using the ROC score.

In this study ordinary least squares regression is applied. This method requires independent predictors and normally distributed residuals. Collinearity in the predictor data set can cause overfitting effects. Despite these limitations, MLM is relatively robust to outliers, can produce good approximations and has been successfully applied in similar studies relating atmospheric teleconnections to drought indicators (Mishra and Singh, 2010), for example in the prediction of Ethiopian rains (Diro et al.,

5   2008, 2011). The strength is the simplistic and reductionistic approach, employing only a few significant predictors.

Predictor selection is performed by the automated bothways-stepwise selection algorithm. Starting from an empty model (intercept only), predictors are added to a model one at a time and the model quality is calculated. The predictor resulting in the highest model quality is retained and more predictors are added likewise in the following iterations. In bothways-stepwise selection once added predictors can still be removed from the model at a later iteration step. The selection process continues

10   until no addition or removal of predictors leads to an increase in model quality. The measure for model quality has to balance the goodness of fit with model complexity, i.e. the number of model parameters. Two measures are applied: Akaike's information criterion (AIC) and Bayesian information criterion (BIC), the latter resulting in more conservative models.





The models' degree of overfitting is tested by performing leave-one-out cross-validation. The difference by which the cross-validated root mean square error ($RMSE_{CV}$) exceeds the $RMSE$ of the model fit residuals is used as a measure of generalisation properties. Furthermore, forecast uncertainty is estimated based on $RMSE_{CV}$. Models are developed for three sets of input aggregation levels and two information criteria (AIC/BIC), resulting in six models per lead time and station. For each

lead time, the model achieving maximum generalisation properties and minimal error configuration is selected (only the selected models are presented in the results). The selected models contain varying numbers of predictors for which the relative importance is calculated. The contribution to the total explained variance, i.e. the predictor importance, depends on the order of the predictors. Predictor importance is calculated with the R-package "relaimpo" according to Lindeman et al. (1980). Drought probability is calculated under the assumption of a normally distributed forecast error estimated using $RMSE_{CV}$ (Diro et al.,

10  2011).

### 2.5.2 Artificial neural network models

The ANN is trained with the genetic algorithm (ANN-GA) which was successfully applied in forecasting rainfall in Eastern Africa by Mwale et al. (2007). The genetic algorithm employs the processes of population growth for the model learning process. The network is designed with at least three layers, each containing a number of nodes. The nodes contain the data and

are connected to the nodes of the next layer by transformation functions. The first layer is the input layer, where each node represents a predictor variable. The last layer is the output layer which contains the response variable. There can be several hidden layers in between, but in this study the models are set up with one single hidden layer. The number of nodes in the hidden layer is varied over four model setups with increasing complexity containing 3, 5, 7, or 10 nodes. All nodes of one layer are connected to the nodes of the next layer. The nodes are parameterised by so called biases and weights, which define how

the input from other nodes are weighted. The values of a node $j$ in the hidden layer $hidunit_j$ is calculated based on the $N$ input nodes $x_i$ by

$$hidunit_j = \sum_{i=1}^{N} W_{ji} x_i + B_{jo} \tag{2}$$

where $W_{ji}$ are the weights for the input nodes and $B_{jo}$ are the biases of the hidden nodes. Then the $hidunit_j$ value is translated by the non-linear function

$$f(hidunit_j) = \frac{1}{1 + e^{-hidunit_j}} \tag{3}$$

and combined by the weights and biases assigned in the output layer to calculate the prediction value.

The next step is the model learning with the genetic algorithm (GA) to determine the best parametrisation. The learning process starts with a random generation of ANN parameters. The GA is an iterative learning algorithm that regards model parametrisations as chromosomes in a genetic population (here 3000), which is subjected to evolutionary processes as with

every iteration step a new generation is created undergoing mutation and crossover. 15% of chromosomes in the new generation are assigned with random parametrisations. First, chromosomes are ranked by forecast skill, called "fitness". Then, the best 85%



of the chromosomes are retained in the genetic pool. However, these are subjected to mutation and crossover processes. During the mutation process in a small part of the chromosomes (here 5%) some of the weights and biases are mutated, i.e. values are randomised. Thereby, small changes in the skilful chromosomes are triggered, which will be retained for the next generation, if forecast skill is improved. In the crossover process, pairs of chromosomes are chosen from the retained chromosomes, and weights and biases are exchanged at one point of the pair of chromosomes (here, with a crossover rate of 0.6). The crossover makes sure that skilful configurations stay in the population and slowly converge to one solution. The procedure is iterated until the root mean square error (fitness) is smaller than 0.005, or a maximum of 1500 iterations. The ANN-GA method is applied as implemented in the R package "ANN" (Roy-Desrosiers, 2012).

The generalisation properties of the forecast models is evaluated by leave-one-out cross-validation in the same way as with the MLM. The ANN-GA result is deterministic and is transformed to a probabilistic drought forecast with the same approach as with the MLM, by assuming a normal distribution with a standard deviation estimated from the cross-validated $RMSE_{CV}$.

### 2.5.3 Random Forest regression tree models

Regression tree modelling is a multivariate data-driven method and a special version of a decision tree tailored to predict continuous variables. Regression trees are used to predict a single variable based on multiple predictors by performing recursive partitioning on the training data. The result is a regression tree, which classifies the data set into small homogeneous groups, so called leaves. Regression trees are strict data-driven multivariate models able to map non-linearity and interactions between predictors. This is a promising feature particularly for atmospheric and hydrological sciences, but up to now the method is not common in these disciplines. Hall et al. (2011) is one of the rare examples where the forecasting performance of Random Forests and other observation-based methods was evaluated. Random Forest regression tree modelling (RFOR) fulfils several desirable characteristics (Breiman, 2001), including ease of parallelisation, robustness to outliers, fast calculation, internal estimates of error, strength and variable importance. The method extends regression tree modelling by introducing a tree model ensemble (here 500) which can be used to represent forecast uncertainty. The trees are learned in a bagging approach, which reduces the overfitting problem of single regression trees. Every tree is trained with a different data set, created by sampling from the original data set with replacement (in-bag samples). All observations not selected are referred to as "out-of-bag" and are used for validation and estimation of variable importance which is described in section 2.7. The Random Forest models are set up with 500 regression trees that have a minimum final node size of five observations. The implementation of the algorithm in the R package "randomForest" by Liaw and Wiener (2002) is applied. Although Random Forest provides an internal error estimation, leave-one-out cross-validation is also applied for the Random Forest models for the sake of exact comparability with the other approaches.

## 2.6 Model and forecast validation

The wealth of potential predictors, some even showing a weak correlation (p.g. ENSO related predictors), increases the risk for overfitting. Overfitting is the effect, that a model can start fitting the noise contained in the predictor data instead of the signal. Robust statistical learning methods minimize the risk of overfitting. Every model learning algorithm is facing a



tradeoff between model fitting and generalisation. Model validation serves the purpose to identify the models with the best generalisation properties. Comparison of the models' forecasting results is performed using the independent forecasts resulting from a leave-one-out cross-validation (LOO-CV), which results in a more realistic estimation of the real forecast uncertainty. The LOO-CV prediction time series resembles a hindcast series. The deterministic forecast performance of the models is

assessed by the coefficient of determination, which is equivalent to the Nash-Sutcliffe efficiency (NSE).

A drought specific forecast verification was performed with the receiver operating characteristic (ROC). The ROC score assesses the forecasts' skill to distinguish between occurrence and non-occurrence of drought which required probabilistic forecast transformation. A moderate level of drought was tested following the definition of drought below -0.5 (SSI < -0.5). In a ROC analysis, a diagram is constructed that presents the hitrate H in dependency of the probability of detection (POD) for a

range of early warning thresholds. According to Wilks (2006), the first step is the calculation of 2x2 contingency tables $C(I)$ for $I$ warning thresholds with $0 < I < 1$. Applied to a probabilistic drought hindcast series, the hitrate

$$H = \frac{N_{correct}}{N_{drought}} \qquad (4)$$

is calculated from the number of correct forecasts $N_{correct}$ and the total number of occured droughts $N_{drought}$. Consequently, the probability of false detection

$$POD = \frac{N_{false}}{N_{nodrought}} \qquad (5)$$

is calculated from the number of false alarms $N_{false}$ and the number of non-drought events $N_{nodrought}$ (wet or normal). The ROC score is the area under the curve and is used for model comparison. A perfect forecast reaches a ROC score of one and a score of 0.5 has no skill, representing a random forecast. The score is calculated with the R package "verification" (NCAR, 2012). This package employs a method by Mason (2008) who showed that the ROC score can be estimated from the Mann-

Whitney U-statistic. In order to estimate the uncertainty of the ROC score calculation, ROC score confidence intervals (95% level) are estimated by 100-fold bootstrapping of the hindcast series and subsequent ROC score calculation.

## 2.7 Analysis of predictor importance

Predictor importance is analysed for MLM and RFOR. The MLM predictor importance is calculated with the "lmg" method by Lindeman et al. (1980), which estimates a partial coefficient of determination for every predictor. These are affected by the

order and combination of predictors in the model. The lmg method minimizes these effects and gives a robust estimate of the true coefficient. Due to the high number of potential predictors, the analysis is focussed on the following predictor groups: Atlantic, Indian Ocean, ENSO, DMI, NAO, streamflow.

Predictor importance in RFOR models is assessed differently. First of all, it is important to be aware that in regression tree models the interdependency of variables is an essential property of the method. The combinations of variables are of

higher interest than single variable importance only, which results in a different approach for importance analysis. Predictor importance for Random Forest models is based on the out-of-bag classification errors. The out-of-bag samples are randomised one variable after the other and the percentage increase in the prediction error is calculated. The understanding is, that the more the randomization of a predictor causes an increase in prediction error, the more important it is.





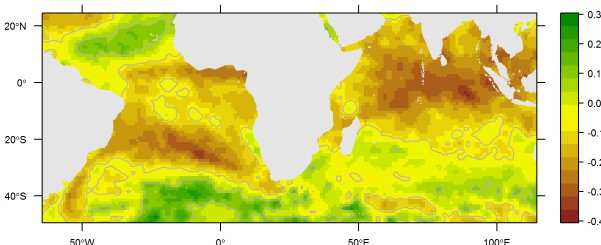

**Figure 4.** Correlation of standardised streamflow (6 months scale) of station Chockwe and sea surface temperature anomalies, grey contours indicate significance at 0.05.

As explained above, predictor importance in MLM and RFOR is calculated differently, thus the indices contain different information. Therefore, the RFOR predictor importance was modified for comparison. Collinearity of predictors can effect the estimation of importance, since predictors might easily replace each other in the regression trees if they have a similar predictive strength. This can cause several effects. On the one hand, the importance per single predictor might be underestimated, if it is

not located at an important position in all regression tree models. On the other hand, in presence of collinearity there would be multiple predictors with underestimated predictor importance. As a consequence, closely related predictors are summarised as relative group importance, calculated as

$$I_g = \frac{\sum_{i=1}^{m} I_{g,i}}{\sum_{i=1}^{p} I_i},$$

which estimates the importance of $g$ predictor groups with $m$ group members and $p$ total predictors. The relative importances

per group can be displayed in a similar manner as the partial coefficients of determination available for MLM allowing for comparison of predictor importance between the methods. When comparing the results, one has to consider the method-specific differences between partial R² and RFOR predictor importance.

## 3   Results and discussion

### 3.1   Identification of customised potential predictors

The list of potential predictors contains several well established climate indices. These cover climate anomalies in the Atlantic, Indian Ocean and Pacific but might not capture the effects in proximity of southern Africa. Therefore, complementary customised climate predictors are deduced from correlation and composite analysis of SSTs in the southern Atlantic and Indian Ocean for drought in the Limpopo basin. SST is correlated with $SSI_{DJFMAM}$ of station Chókwè, which is the largest sub-basin. The Spearman correlation coefficient ranges from -0.36 to 0.26 with a median of -0.08 (Figure 4). The correlations can

be considered low but a large share of the correlations is still significant given a large sample size of 724 observations. Negative correlations are found in the northern Indian Ocean and the Atlantic from 10°N to 30°S. They indicate that warm anomalies



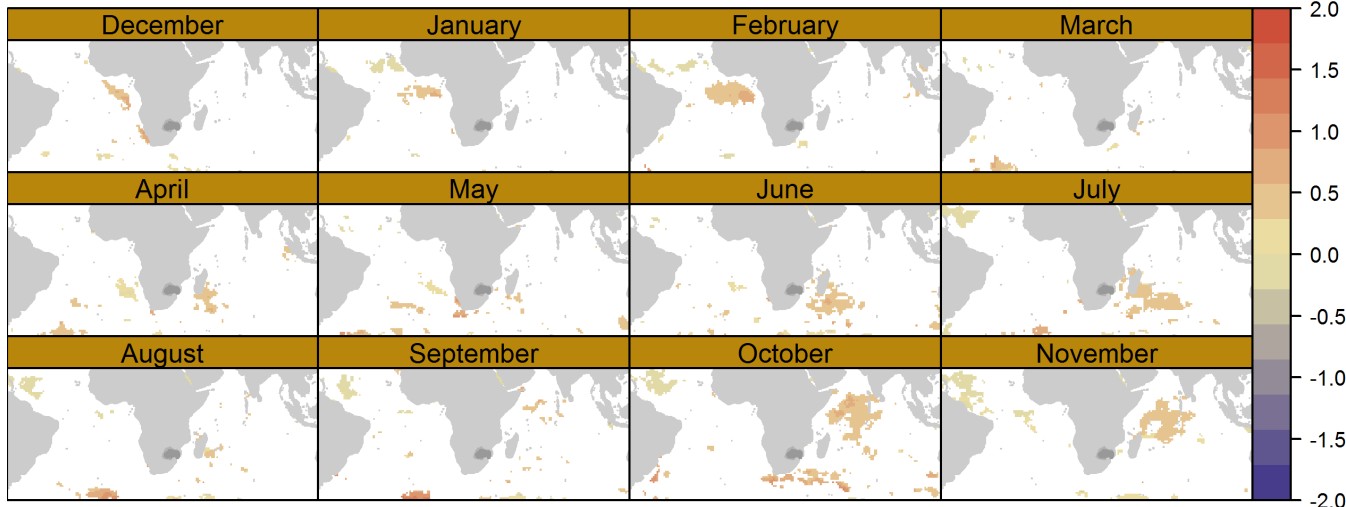

**Figure 5.** Composites: Anomalies of sea surface temperature preceding drought during DJFMAM, significant anomalies presented only.

are related to drought in the Chokwé signal. Correlations are strong in the Gulf of Guinea and the central southern Atlantic. Positive correlations are located in the southern ocean regions dominated by the circumpolar current. South of Cape Horn small scale random patterns are found. At the Namibian and Angolan shoreline a relatively thin zone exhibits positive correlations as well, which might be attributed to upwelling cold water from the Benguēla current in the region.

The composites analysis for conditions preceding $SSI_{DJFMAM}$ drought shows more restricted regions with significant anomalies (Figure 5). Droughts are associated with positive anomalies in the north-western Indian Ocean during the preceding October and November. During June and July positive anomalies occur in the south-eastern region south of Madagascar. In the Atlantic positive anomalies are significant in the Gulf of Guinea at longer lead times of 10 to 12 months. In October positive anomalies also appear south of the African continent. Similar to the correlation analysis these show a high small scale

variability. This ocean region is characterised by a complex system of currents with upwelling cold water of the Benguēla current in the West and the Agulhas warm water current in the East which collides with the South Atlantic current. The anomalies in the mixing region of the South Atlantic current and the Agulhas current are very small and might be related to warm or cold water eddies (see Figure 1), which form under the special conditions of the two mixing currents (Peterson and Stramma, 1991). Due to the small extent of the anomalies and the chaotic nature of the eddy formation these anomalies are

not included as predictor. Nevertheless, the currents themselves are represented by other Ocean regions in the Indian Ocean (predictor "Agu") and the southern Atlantic (predictors "SWAtl", "SEAtl", "BC").

    As a result of the correlation and composites analysis, in total ten ocean regions are defined (see Table 4 and Figure 6) and potential SST predictor indices are calculated with three aggregation levels (1, 3, 6 months), resulting in 30 customised SST indices. The total set of potential predictors comprises 55 variables. 16 of these are well known climate indices, of which



**Table 4.** Potential customised predictors from ocean regions teleconnected to drought in the Limpopo basin. Region selection is based on correlation and composites analysis. Coordinates indicate extents of the polygons (minimum, maximum).

| Ocean region | Latitude | Longitude | Abbreviation |
|---|---|---|---|
| *Atlantic* | | | |
| Benguela current | -34, -6 | -12, 13 | BC |
| Southern African Coast | -38, -20 | 8, 25 | SACoast |
| Southwestern | -48, -40 | -40, -15 | SWAtl |
| Southeastern | -48, -40 | -15, 15 | SEAtl |
| Guinea Gulf | -5, 7 | -25, 12 | GuiGulf |
| Guinea coast | 7, 20 | -40, -10 | GuiCoast |
| *Indian Ocean* | | | |
| eastern equatorial | -18, 2 | 75, 120 | eeqIO |
| western equatorial | -10, 5 | 40, 60 | weqIO |
| Agulhas current | -35, -20 | 28, 45 | Agu |
| South of Madagascar | -37, -27 | 45, 60 | SMad |

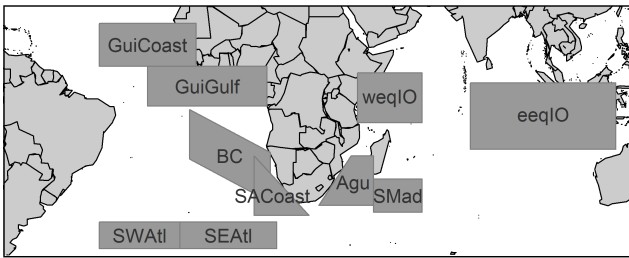

**Figure 6.** Regions of customised SST indices. Predictor region abbreviations are listed in Table 4.

14 are related to El-Niño and SOI. In addition to NAO, the influence of the Atlantic region is represented by 18 customised predictors (6 regions by 3 aggregation levels). The Indian Ocean, represented by the climate index DMI, is complemented by 12 additional customised predictors (4 regions and 3 aggregations levels). Furthermore, there are three predictors representing the antecedent catchment conditions in the form of current standardised streamflow at aggregation levels of 1, 3 and 6 months.

5  **3.2  Intermodel comparison of predictor selection and importance**

MLM and ANN models consist of specifically reduced predictors sets, whereas RFOR relies on the complete predictor set. Therefore predictor selection frequencies are presented for MLM only. Predictor importance is compared for MLM and RFOR,





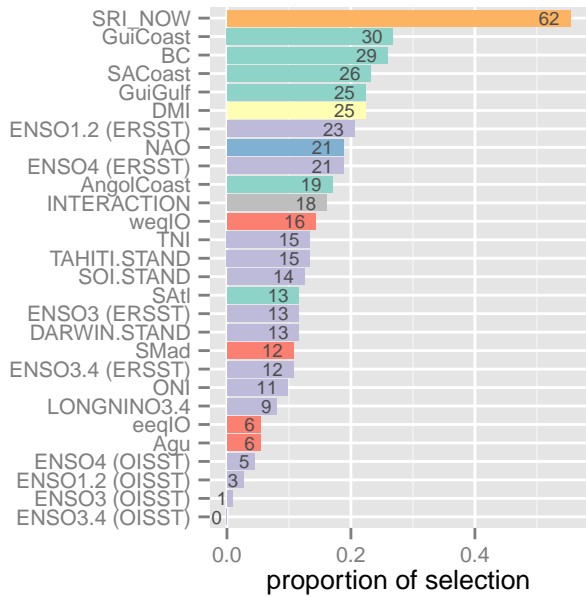

**Figure 7.** Predictor proportion of selection out of 112 (7 lead times for 16 stations) multiple linear models, with absolute number as text. Bars are coloured according to groups: Customised Atlantic (green), DMI (yellow), ENSO (purple), customised Indian Ocean (red), NAO (blue), streamflow (orange), interactions of selected predictors (grey).

for which different estimators of predictor importance exist: Partial coefficient of determination and RFOR predictor importance.

The proportion of selection of the predictors in the MLM models (Figure 7) shows which predictors are frequently part of the final MLM (and ANN) models. Antecedent streamflow is selected with highest frequency (Figure 7) followed by several

customised indices of the Atlantic and DMI. The first of the El-Niño related parameters (ENSO1.2) is in seventh place. Every ENSO related predictor has a selection frequency of less than 0.21, which is rather low given the relevance of ENSO in the region. The indices are based on different SST ocean regions (ENSO 1.2, 3, 3.4) or are calculated based on SLP (SOI), however, they are correlated and the indexes might easily replace each other in different selection runs. As a result, the proportion of selection might be low for specific ENSO indices, but not for the ENSO anomaly altogether. Furthermore, ENSO is correlated

with DMI and Darwin SLP, which even was identified as superior to ENSO for drought prediction in Zimbabwe by Manatsa et al. (2007). Here, Darwin SLP is only selected in 12% of the models which does not support that finding. Additional ENSO related predictors are tested originating from two different data sets: ERSST and OISST. , ERSST provides longer time series, whereas the OISST data set is preferred to ERSST qualitatively due to the inclusion of new types of SST observations. This, however, does not result in a preferred selection of OISST ENSO indices, which are selected only rarely.

In fact, over all stations and lead times ENSO related predictors form the most important group of SST predictors. The group includes ENSO indices but also SOI, as well as Darwin and Tahiti SLP. Predictors from this group are selected in 80% of the





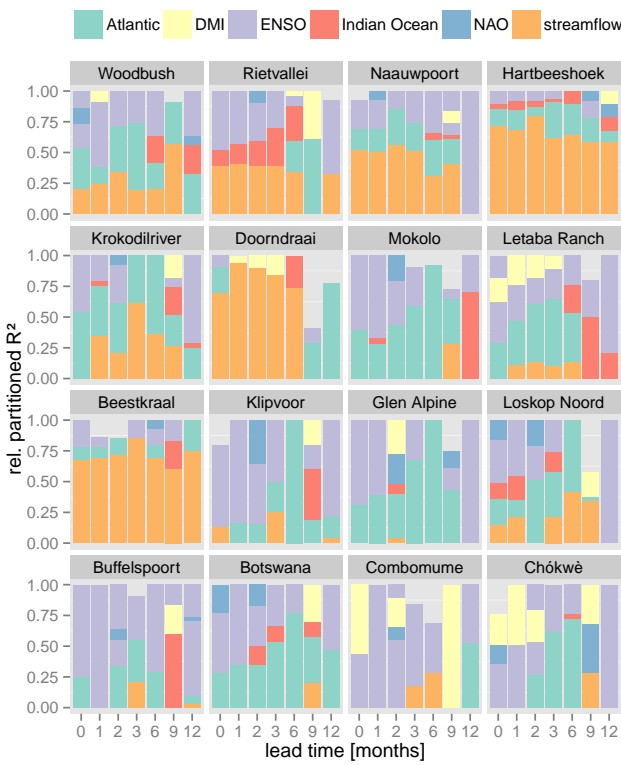

**Figure 8.** Relative partial R² by the predictors in the MLMs summarised as follows: Antecedent streamflow conditions, Indian Ocean dipole mode index (DMI), El Niño-southern oscillation indices (ENSO), north Atlantic oscillation index (NAO) and the customised indices of the southern hemispheric Atlantic and the Indian Ocean. Parameter interactions are excluded from the plot and account for the white gaps. Stations are ordered by catchment area from small (top left) to large (bottom right).

linear models (Figure 8). The ENSO predictors' contribution to overall explained variability differs widely between models but the median is rather high with 41.2%. However, in contrast to the majority of stations, the stations Haartbeeshoek, Beestkraal and Doorndrai show only a weak or even absent effect of ENSO predictors (Figure 8).

The customised predictors for the Atlantic are incorporated in many linear models (73%) and the median of the relative partial coefficients of determination reaches 31.8%, which is only slightly below the contribution of the ENSO indices. The contribution is particularly strong at the stations Glen Alpine and Botswana. Besides El-Nino and the Indian Ocean, the Atlantic is also described as an important factor for southern African rainfall by Reason et al. (2006). In their study the predictability of rainfall is attributed to the influence of the Benguela current and the SST of the south-eastern Atlantic, which is related to the South Atlantic current. In addition, our results indicate a connection to the SST of the Guinean Gulf (GuiGulf) and equatorial Atlantic (GuiCoast) in Figure 7. The NAO index is also included in the MLM models, but only at a rate of 18%. The contribution of NAO to the explained variance is 13% (median).





Antecedent streamflow (SRI_NOW) is selected in 55% of the linear models and is very important for many of them. In half of the models, in which antecedent streamflow is included, the predictor contributes at least 42% of the explained variance (median rel. part. $R^2$). The importance of antecedent catchment state is supported by van Dijk et al. (2013), who found that initial conditions provided most skill opposed to meteorological forcing in a forecasting experiment with a global ensemble stream-
flow prediction system. Antecedent streamflow is particularly prominent in smaller catchments (Figure 8). In statistical models, antecedent streamflow is a common predictor, which exploits signal autocorrelation that is caused by the delayed rainfall-runoff response in the hydrological system (Robertson and Wang, 2012). As a parameter representing catchment memory and other autocorrelation properties, it could be expected that the importance of the predictor decreases with higher lead times. This effect is observed at stations Hartbeesthoek, Doorndrai, Krokodilriver and Nauwpoort, but the decrease is not strong. The most
obvious effect is present for lead time 12, where antecedent streamflow is selected only in four of 16 stations.

The Dipole mode index (DMI) of the Indian Ocean is selected less often. 22% of all models include the index as predictor and its median share in the models' explained variance is 20.1%. This is of particular interest given the selection rate of the customised indices of the Indian Ocean, which reached 30% with a median relative partitioned $R^2$ of 0.15. The Indian Ocean predictors are selected particularly at longer lead times. In short, DMI is seldomly selected, but has a comparatively high
importance in the models.

The predictor importance of the MLM models differ strongly between lead times and stations. Several cases exhibit very different parameter selections than foregoing lead times, for example lead time nine of station Rietvalley. These cases result in an impression of randomness in predictor selection, which indicates that these observations are statistical artefacts. One possible reason might be that, on the one hand, these statistical artefacts could occur when selection is performed under
nonideal conditions, e.g. collinearity. On the other hand, different predictor configurations might lead to very similar AIC/BIC values, but only the model with the highest value is chosen. As a result, the estimated predictor importance for the MLM models is highly specific to the selected models and can be inconsistent between lead times.

In contrast, the results of the RFOR models can provide a more general picture, as they always include all predictors and use a randomisation process to estimate predictor importance, shown in Figure 9. For the ease of comparability, Figure 9 is designed
similarly to the relative partial $R^2$ of the MLMs presented in Figure 8, but it is important to note that the importance measures are different (see section 2.7). RFOR predictor importance shows the sensitivity of the model error to the individual predictors. RFOR produces a more even pattern of predictor importance than MLM. This is caused by the fact that RFOR encompasses all predictors. The randomized RFOR ensembles are then compared to single MLM realisations, which are bound to one specific selection.

The RFOR predictor importance shows four main differences and two confirming features in comparison to the MLM results. First, the Atlantic has a more constant and stronger importance. Second, in contrast to the MLM results a stronger effect by the lead time is observed, for example at station Doorndraai, where the importance of streamflow decreases with higher lead times (Figure 9). And at several stations the importance of the Atlantic ocean predictors increases from 0 to 6 months lead time and drops thereafter, which is also observed in the MLM models. Third, the predictors from the Indian Ocean are more important





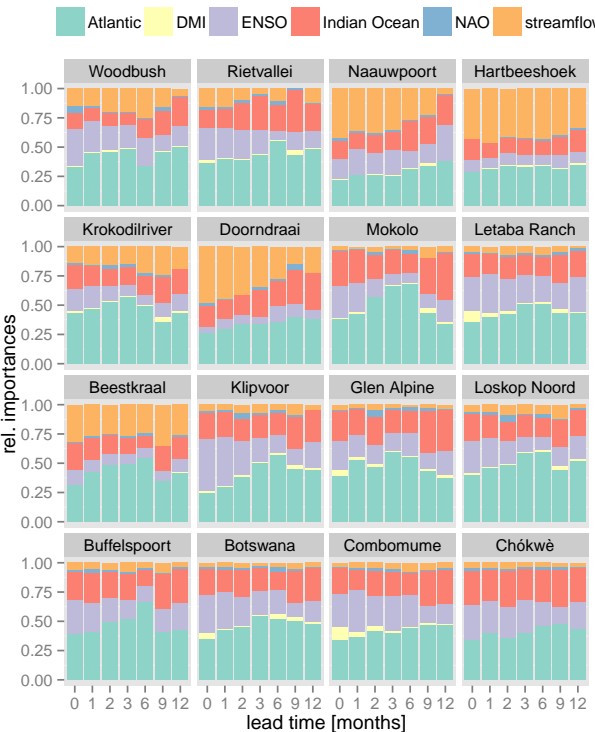

**Figure 9.** Relative importance of predictors in the RFOR models grouped as follows: Antecedent streamflow conditions, Indian Ocean dipole mode index (DMI), El Niño-southern oscillation indices (ENSO), north Atlantic oscillation index (NAO) and the customised indices of the southern hemispheric Atlantic and the Indian Ocean. Stations are ordered increasingly by catchment area from small (top left) to large (bottom right).

at all stations and are more constant over all lead times. Fourth, ENSO predictors are less important compared to the MLM models, where they are the dominant predictor group.

The RFOR results confirm the major relevance of antecedent streamflow at different lead times, and produces a very similar pattern compared to relative partial $R^2$ of the MLMs, where streamflow is a strong predictor at stations with smaller catchment

5  areas. Furthermore, the effect of decreasing importance of streamflow with longer lead times was strongest at the stations Naauwpoort and Doorndrai. The results also confirm the lower value of NAO and DMI. Overall, the customised SST indices in the Indian Ocean are more emphasized and more persistant for all stations and lead times compared to the MLM results. This might be an effect of the forced selection of only a single final MLM, that causes the Indian Ocean indices to be dropped from some models. However, it might also indicate that indices in the Indian Ocean in particular have a conditional relationship with

10  other indices, which could only be represented by RFOR and not MLM. The low forecasting skill achieved by RFOR does not encourage further investigation in this matter.



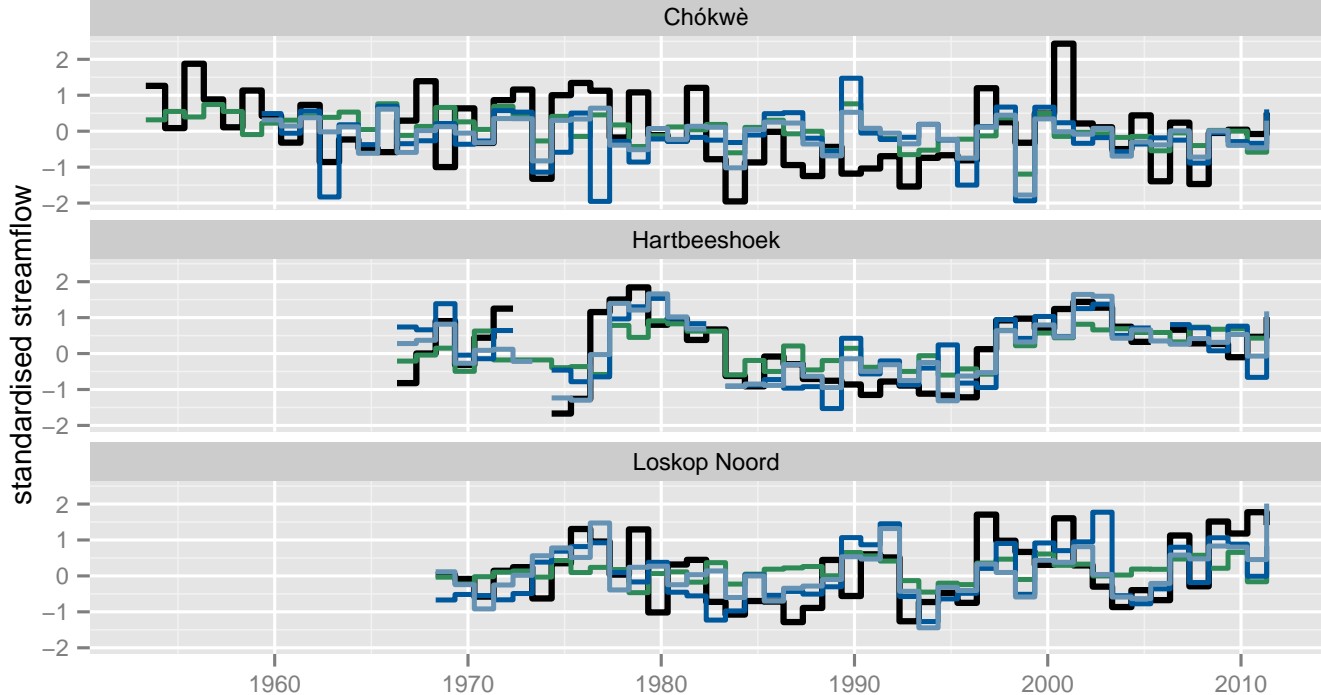

**Figure 10.** Hindcasted time series of DJFMAM forecast models at lead time one (November forecast) of stations Chókwè with low, Hartbeeshoek with good and Loskop Noord with medium skill. The presented time series are the crossvalidated, thus independent, results of the models MLM (light blue), ANN (dark blue) and RFOR (green). Observed standardised streamflow is shown by the underlying black line.

In summary, the study shows that customised SST indices can contribute substantially and even outperform global climate indices in predictor importance. It is well known that ENSO has an impact on drought occurrence in southern Africa, but the strength of the relationship has been questioned since other indices like the Darwin SLP or DMI also exhibits a strong - and supposedly stronger - influence as suggested by Manatsa et al. (2007). Distinguishing the importance of these correlated

5   predictors is difficult but our results do not stengthen that finding. In our study DMI and Darwin do not exceed the importance achieved by ENSO. Instead, specifically customised indices in the Atlantic and Indian Ocean show good potential in statistical drought forecasting. There is no doubt about the strong effect of ENSO on global circulation patterns. Still, our study suggests that predictability studies are advised to create customised indices that refer to the drought specific SST anomalies in the region surrounding the area of interest to supplement the global indices. By doing so, it is more likely to capture all factors leading to

10  drought events.



### 3.3 Forecast skill for drought early warning

The forecast skills vary widely between stations, models, lead times and time periods. For example, the forecast of the largest subcatchment (station Chókwè) achieves low skill (see Figure 10). A few drought events are forecasted well at lead time one (e.g. 1974), but often event magnitudes are not met (after 2000) or overestimated (several events between 1977 and 2000).

The forecast of Hartbeeshoek at the same lead time presents a more promising picture, where models are able to represent the long term variability of the streamflow but the interannual variation is overestimated by the models. The forecast at the station Loskop Noord is of medium skill where, for example, the dry period during the mid nineties is forecasted but several other extreme events are not. These three examples give an impression of the range of forecast results achieved by a lead time of 1 month.

The forecast skill can be assessed in many ways (Wilks, 2006), both deterministically and probabilistically. In this study the skill of the different forecast models is deterministically evaluated with the leave-one-out cross-validation Nash-Sutcliffe model efficiency (NSE), and probabilistically with the ROC score. The former measures the accuracy in forecasting the exact deterministic SSI value, whereas the latter assesses the discriminative skill of a probabilistic drought forecast in early warning mode.

An analysis of the deterministic skill for all stations and lead times, using LOO-CV NSE, reveals that MLM produces the most robust forecasts and achieves the highest forecast skills with a maximum of 0.73 and a median of 0.30 (Figure 11). The maximum skill reached by the ANN is a little bit lower with 0.61 but the median is very low with 0.03 which is caused by the absent skill at many stations. The ANN forecasts only achieved considerable skill at the stations Nauwpoort, Hartbeeshoek, Krokodilriver, and a few more cases. The ANN models strongly suffer from overfitting. Regardless of the number of hidden 20 neurons, the difference between fit and crossvalidated error is higher than for MLM (data not shown). Employing ANN with the predictors selected for the MLM does not lead to improved forecasts skill in this study.

At stations Naauwpoort and Hartbeeshoek the predictability by all model systems is highest. The model skill shows strong inter-station variability, which is not unusual in streamflow forecasting (Robertson and Wang, 2012). While MLM achieves skills like in Naauwpoort and Hartbeeshoek at a few more stations, ANN and RFOR only rarely reach that level. The skill is 25 highest in the smaller subbasins (upper two rows of Figure 11) and lowest in the bigger catchments (lower two rows of Figure 11).

The skill of the probabilistic drought forecasts is analysed with the ROC score presented in Figure 12. The forecasts have more skill than a climatological forecast, i.e. ROC > 0.5, at almost all stations and lead times. As in the deterministic evaluation, skill levels vary strongly between stations. Prediction skill decreases with higher lead times at most stations. The reduction 30 in skill is most pronounced for the longest lead times 9 and 12 months. Exceptions to this are the stations Beestkraal and Hartbeeshoek having almost constant skill at all lead times. The skill at stations Buffelspoort and Chókwè is generally on a lower level and exhibits an unusual pattern where longer lead times have higher skill.

The three statistical methods achieve different median ROC scores. When models are ranked per station and lead time, the majority of first ranks is achieved by MLM models, most second ranks are ANN and most third place ranks are RFOR models





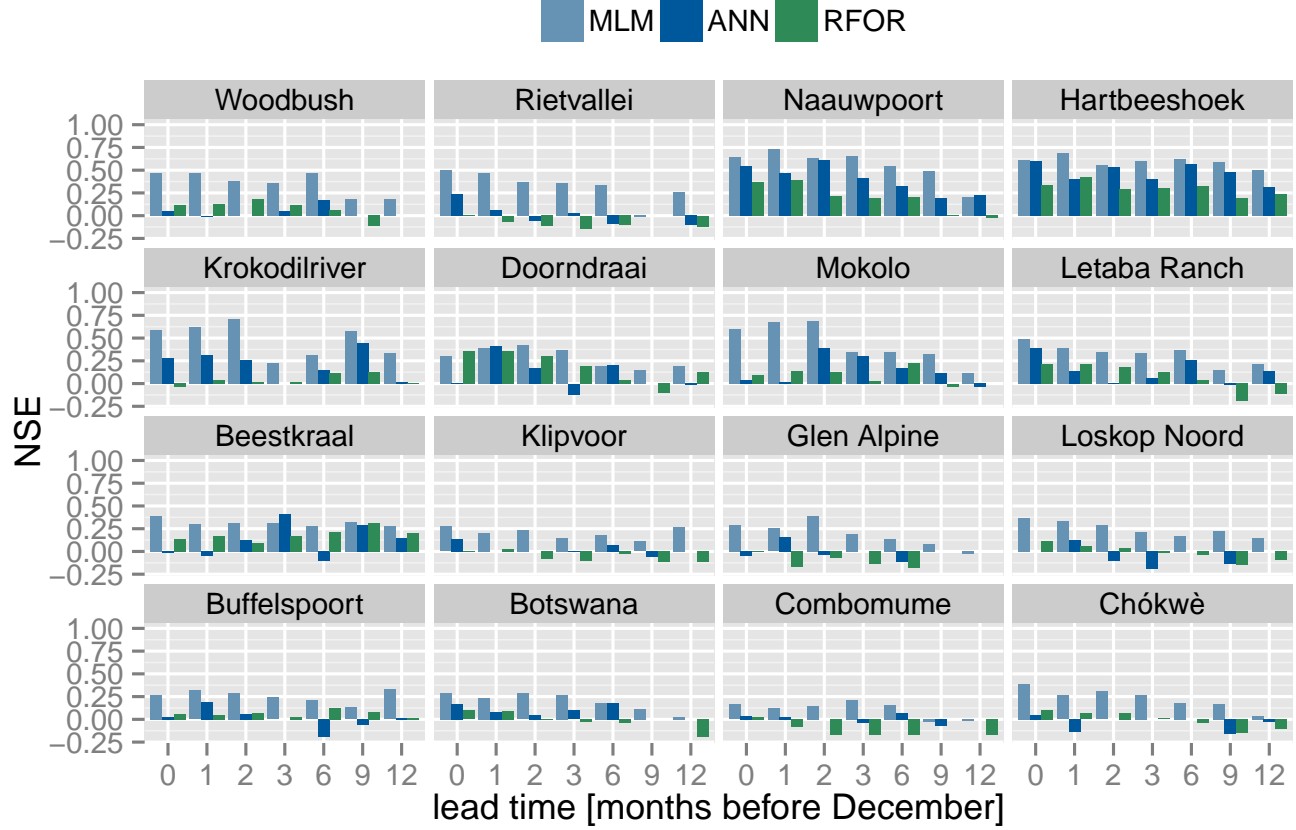

**Figure 11.** Validation of DJFMAM drought forecasts with LOO-CV Nash-Sutcliffe model efficiency: Multiple linear models (MLM, bright blue), artificial neural networks (ANN, dark blue) and Random Forest models (RFOR, green). Stations are ordered increasingly by catchment area from small (top left) to large (bottom right).

(73%, 63% and 63% of the models per rank, respectively). MLM often reaches the highest skill and is therefore ranked in first place, but it has to be noted that often the differences are minor. Also, there are a few instances where this pattern does not hold. For example, RFOR is much more skilful in forecasting station Chókwè at lead times 0, 1 and 2. These results are similar to the strong inter-station variability found by Robertson and Wang (2012) in a streamflow forecast study for Australian catchments.

5    Another interesting feature of the results is the variability of the error bars associated to the ROC scores. The error bars are derived by resampling the hindcast series and indicate the influence of individual observations on the robustness of the model predictions. It can be seen that for stations with generally good skill the errors are also small, while for the stations with lower skill the errors tend to be larger. Large errors indicate that a few observations have strong influence on the forecast skill, implying that drought forecasts in these basins are generally difficult with the presented models.





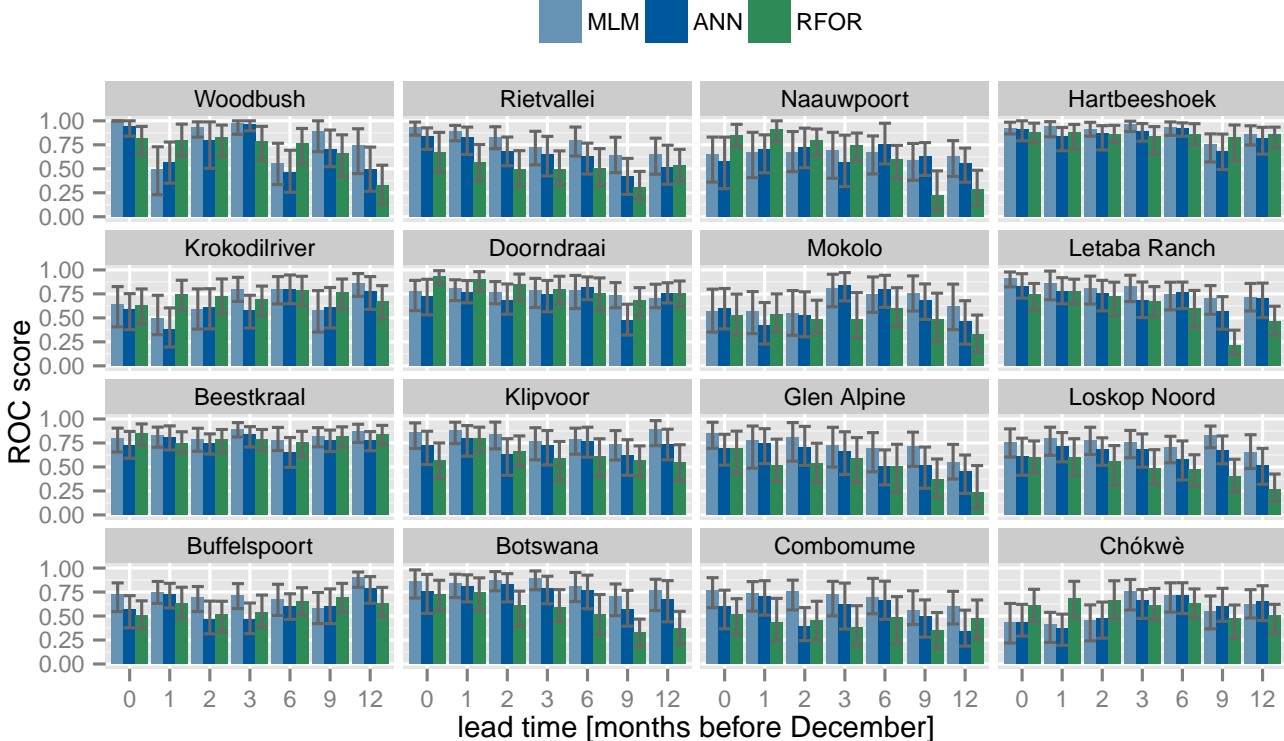

**Figure 12.** Validation of DJFMAM drought forecasts with ROC scores: Artificial neural networks (ANN, dark blue), multiple linear models (MLM, bright blue) and Random Forest models (RFOR, green). Bars indicate median ROC scores, error bars show bootstrapped 0.95 confidence intervals of the scores. Stations are ordered increasingly by catchment area from small (top left) to large (bottom right).

Relating the ROC skill with the selected predictors reveals that stations with high ROC scores coincide with a high influence of antecendent streamflow. This is an indication that catchment conditions play an important role for both the development and predictability of droughts. Interestingly, our results show a much higher skill in forecasting droughts in smaller catchments. This might be explained by the degree of human interferences. A plausible hypothesis is that the degree of human interference

5    with streamflow is lower in smaller catchments. The low prediction skill in large catchments might be related to the complexity of the large catchments, where many dams, irrigation schemes and groundwater extractions interact and thus cause lower predictability with the adopted methods. Another reason for the better forecast skill for smaller catchments might be a regional bias, since most of the smaller head water catchments are located in the South of the Limpopo basin. A definite answer must be left for further analyses with scope on the role of human interferences in the Limpopo basin.

10    In order to evaluate the forecast skill of the proposed models in relation to other approaches, a benchmark forecasting system would be necessary, but is not available. However, Dutra et al. (2013) published seasonal forecasts of SPI-6 for the Limpopo





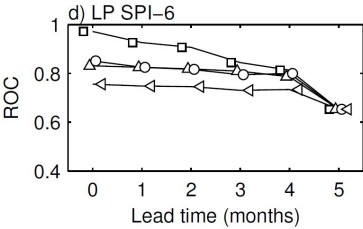

**Figure 13.** ROC scores of seasonal forecasts of SPI-6 of the Limpopo basin by Dutra et al. (2013). Lead times have been defined differently in the following way: The lead time represents the number of months before the last of the 6 months of SPI-6, meaning lead time zero would correspond to a forecast of $SPI_{DJFMAM}$ in May, hence lead time five would be equivalent to lead time zero in this study. The lines indicate different precipitation datasets. (Source: Figure 11d from Dutra et al. (2013))

(Chókwè subbasin) but did not present ROC scores specific for a $SPI_{DJFMAM}$ forecast in December and earlier. The studies' lead times correspond to lead times -1 to -5 according to the definition used here (see table 3) and result in forecasts issued in January to May. The forecast skill (ROC scores) in Dutra et al. (2013) decreases strongly approaching 0.5 (no skill) at a lead time corresponding to January for the $SPI_{DJFMAM}$ forecast (Figure 13). In comparison, the results presented here for

hydrological drought forecasting exceed the skills of the highest presented lead time in Dutra et al. (2013), so does station Chókwè which covers the same area. It has to be noted, though, that Dutra et al. (2013) present continuous forecasts for all months of a year, and intraannual variation of predictability is not shown.

    In summary, there is reason to regard the seasonal prediction of hydrological droughts in the Limpopo basin as challenging. NSE is on a low level at most stations. In addition, the skill scores of the MLM models sometimes appear random despite the

thorough modelling design. For example, the selected predictor combinations can change completely from one lead time to the other. Although this might be an artefact caused by the collinearity of the predictors and the crisp selection of the predictors (section 3.2), these results indicate a generally low predictability of $SSI_{DJFMAM}$ at the lead times presented here. The predictability as tested in this study can build on meteorological influences, for example sequences of weather patterns, represented by SST anomalies. The hydrological influences, for example catchment memory effects, are represented by antecedent

streamflow. The signal contained in the streamflow series which is predictable by the adopted methods seems low. The forecast uncertainty given the long lead times is aleatory. So is the uncertainty of the streamflow measurement. Yet, a considerable part of the uncertainty is epistemic, i.e. it is likely to be reduced by model improvements or additional data. There are several factors unaccounted for, that could reduce the epistemic error. Examples are the introduction of anthropogenic interference with streamflow (abstraction, storage) or the introduction of further hydrological and meteorological parameters. However, despite

the generally low forecast skill, the importance of predictors presented in this study indicates the governing climate conditions for droughts in the Limpopo basin.





## 4 Conclusions

This study presents the predictability of hydrological droughts in the Limpopo basin. It transfers methodologies to hydrology that have been used predominantly in meteorology. The results show that hydrological drought in the Limpopo can be predicted based on SST anomalies, although the predictability varies between catchments and lead times. Seasonal forecasting is a

demanding task in a catchment with high anthropogenic interference with the hydrological processes. Nevertheless, seasonal to annual predictability is still present in the streamflow signal which is shown by forecasting skills that exceed climatology. This study has four main findings:

First, a seasonal forecasting approach proves successful, which forecasts standardised streamflow with statistical methods based on climate indices, sea surface temperature (SST) teleconnections and antecedent streamflow. Streamflow is expressed

in terms of a standardised streamflow index (SSI), which enables a better identification and a common definition of droughts in basins of different characteristics and size. Standardised indices are less common in hydrology than in meteorology, but we recommend their use in hydrological drought studies.

Second, the most important climate predictors are ENSO-related and customised drought predictors in the Atlantic. In addition to the climate indicators, antecedent streamflow as a proxy for catchment state proves to be another important predictor.

Regarding antecedent catchment conditions, the catchments are separated in two groups: one group with a strong importance of antecedent streamflow and comparatively high forecast skill and one with low (or absent) influence and lower forecast skill. The reason for that pronounced effect could not be answered in this study and must be left for future work. Possible causes are the degree of human interference in the catchments, which is likely to be lower in smaller catchments, or a regional bias in the predictability caused by climatology. However, the importance of antecedent streamflow underlines the relevance of catchment

condition for hydrological drought prediction.

Third, the best forecasting skill within this study is achieved with multiple linear models. Based on the results of the cross-validation, linear relationships are more robust than the non-linear capabilities of artificial neural networks and Random Forest models. ANN and RFOR are likely to suffer from overfitting of the models, which is in turn a consequence of the limited data set used for training. These results are specific to this region and dataset, but they underline the necessity to benchmark more

advanced methods with simple methods.

Forth, in order to determine a forecasts value for early warning, a thorough forecast system verification is imperative. Verification must incorporate both the deterministic properties using e.g. the Nash-Sutcliffe Efficiency (NSE) and the probabilistic properties using skill scores like ROC. The deterministic forecast skill shown by NSE is low in many stations, but the analysis of the discrimination properties of the probabilistic forecast shows that the forecasts still exceeds a pure climatological forecast

and therefore should not be neglected. Up to now, climatology as benchmark is fine, since water management in the basin is typically relying on it instead of seasonal forecasting as a basis for decision making.

This study shows that hydrological drought can be predicted using statistical methods and teleconnection indices and catchment condition as parameters. The method can be applied in places with available observed streamflow. It is useful when station specific forecasts have value for water management and decision makers. Its simplicity and low computational demand make



it even adoptable as a customised forecast system at an end user level for dam or subbasin management. Thus, it is suited for a bottom-up early warning system at the local level, where decisions are set in action.

*Acknowledgements.* We thank the European Union for funding the work under the EU-FP7 project DEWFORA (Improved Drought Early Warning and Forecasting to strengthen preparedness and adaptation to droughts in Africa).

5      In peculiar we thank very much Lars Gerlitz, Stefan Lüdtke and Viola Kelle for reviews, discussion and support during the preparation of this publication.



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
