# Peer review of "Seasonal forecasting of hydrological drought in the Limpopo basin: A comparison of statistical methods."

_Hydrology and Earth System Sciences, 2016_

## Referee Comment (RC1) · Anonymous Referee #1 · 3 Apr 2016

This manuscript investigates the capability of Multiple Linear Regression (MLM), Artificial Neural Network (ANN) in conjunction with Genetic Algorithm (GA) and Regression Tree to forecast drought in Limpopo basin based on large scale and customized scale climate factors, mostly SST based. The manuscript is well presented and the method and discussion is clear. The results indicate that, with the use of customized SST based factors as predictors, the MLM model is capable of forecasting the drought in the region with a higher forecast skill compared to the other two methods. One of the main contributions of this manuscript is revealing how the customized climate factors have increased the forecast skill compared to the large scale climate indices such as ENSO. Comments: Page 7, line 3: "The standardised streamflow indices (SSI) are calculated for each station at the scale of 6 months. SSI6 May of May at that scale covers the desired main runoff period from December to May, henceforth named SSIDJFMAM (Figure 2)".

When discussing the SSI, it is not clear if the SSI is a single value (averaged or summation?) for the months Dec-May or each month has its own SSI value. I would assume that there is only one SSI value for Dec-May, in that case Figure 2 is showing the Box-Plots of Monthly streamflow and not the SSI. I don't see the use of Figure 2 in this manuscript in relation to SSI.

Page 13, line 2: "Therefore, the RFOR predictor importance was modified for comparison." How was it modified?

Suggestion: As ANN is not bound by any linear assumptions (as opposed to MLM), the use of the MLM predictors which were selected based on Pearson correlation (a linear technique) and relying on MLM stepwise predictor selection has limited the performance of ANN in this study. I suggest that in the future studies, the authors do not bound ANN to limited linear selection of inputs (predictors) and investigate a wider range of inputs using either a simple method of trial and error with ANN or more complicated methods such as mutual information or genetic algorithm to select ANN's inputs.

Please also note the supplement to this comment:
http://www.hydrol-earth-syst-sci-discuss.net/hess-2016-4/hess-2016-4-RC1-supplement.pdf

---

## Referee Comment (RC2) · Anonymous Referee #2 · 26 Apr 2016

This manuscript presents three methods for forecasting seasonal hydrological droughts in the Limpopo basin. All three are statistical forecasting methods, including multi linear regression, artificial neural networks, and random forest trees. The results presented are interesting. I think the development of the statistical methods themselves is of interest, though this has also been shown to be successful in several other publications. I do think that the finding of skill, albeit small, out to lead times of up to 12 months is quite remarkable. Also of scientific interest is the development of custom indicators in the Indian and Atlantic oceans, rather than the authors relying solely on the use of the standard set of indicators. Overall the manuscript is well written, though there is some room for improvements. This is particularly so for the conclusions where the writing

style reverts to very short staccato sentences. I would also like to ask the authors to carefully revise how they refer to the various indices, which is somewhat confusing at times. What confused me is that there are standard indices in both Atlantic and Indian Ocean, but also customized indices. The latter are then referred to as Atlantic and Indian Ocean only. I would propose that the authors revise this and always precede the customised indices with the word "Custom" or something like that. That would help clarify somewhat to my mind.

Overall the figures in the manuscript could be made a little larger to enhance visibility/interpretation. There are some very small figures, and at times the figures are not easy to read (e.g Figure 10 could be improved by plotting the thick black line differently).

My main comment on the paper is the influence of the dams within the catchment is not well explored. In some places the authors elude to the presence of dams, and also include details as to their total volume compared to the average annual volume that enters the dam. This shows that for some of the stations the anthropogenic influence is substantial. In many cases there is more storage than there is annual volume, such as is the case for Nauwpoort. And yet this is one of the two stations that are reported to have the highest skill (together with Hartbeeshoek, which has no upstream, dams). This is surprising. This is also linked to one of the findings of the authors that the predictability of the smaller catchments is better than for the larger catchments. This is an interesting conclusion because it is somewhat counterintuitive, because as the authors note, the lower skill in large catchments may be due to the anthropogenic influences. However, these catchments are really very small. This would mean that the skill found cannot be due to the persistence of the catchment initial conditions.

The last overall comment I have is on the selection of the customised indices. The authors note that these were selected over a large area. However, I can imagine that there is a trade-off between the large area and the ability to find significant differences/anomalies. I would expect that as the area gets larger, the detection of anomalies gets smaller. Perhaps the authors could comment on this. The manuscript

is oriented somewhat at hydroinfromatics techniques, and could be considered more suitable for e.g. The Journal of Hydroinformatics. I think to strengthen its appeal to the HESS community it would be good to include some hydrological reasoning why specific parameters are selected in the statistical models.

Specific comments: P1L4: assessed using statistical P1L9: as a proxy P1L15: warning, the models P2L5: which have severe P2L8: which may even P2L9: regarded as being highly affected P2L11: to studies that found P2L10-12: There is some discussion on the climate. I am not sure these comments are entirely relevant to this manuscript. P3L1: Atmospheric circulation processes have ... P3L6: it extends from the ocean P3L13: by the chaotic P3L26: These are particularly P3L33: The skill of the forecasting P3L33: The authors refer to the DJFMAM forecast. That is clear that this spans the wet season. But is this a single value, or is there a forecast for each month. Perhaps I missed it, but it may be good to clarify in the text what a forecast actually contains in terms of parameters and time steps. P5L3: There is some discussion on extracting the catchment areas. Why are these relevant other than to be included in the table describing the catchments. P5L12: event anomalies P6 Table 1: It may be useful to include the year in which the dam was built, or at least the main dam building period in the Limpopo. This can help interpret possible issues of stationarity in the time series. P7L20: with df = N-2 degrees of freedom P7L23: The region outlines P7L23: generously, so as to P8Table3: It is not so clear what the aggregation period of the streamflow indices is. Are these for one month? Or rather is the SSIDJMAM the aggregated streamflow index across the whole wet season. This should all be clarified a bit better. P9L2: linear regression is applied to estimate the values of parameters Bo to Bp. P9L10: until the addition or removal does not lead to an increase in model quality. P10L20: The hidunitj variable is somewhat long and should be avoided. Perhaps introduce something simpler, such as H, and explain it well. P11: I was not so clear how the forecast skill of the ANN is expressed, and if that is commensurate with how it is expressed for the criteria used to establish the MLM parameters. Please ensure that these are well defined, and that that if there are differences explain why the calibrated

models may then be compared. P11L22: The trees are trained P11L26: I am not sure what is meant by the final node size. P11: Overall the description of the Random Forecast Trees is difficult to follow for those not familiar. What are the 500 regression trees? What is the minimum final node size? I think the majority of the readers of HESS will not be familiar with this technique. ANN is more familiar I think. The authors use quiet a lot of jargon such as "bagging" etc. I would be very helpful if they provide a simple explanation of this technique and how a forecast is actually derived. P12L1: model over fitting (I changed this but please check the context) P1211: It is suggested that 2x2 contingency tables can be used only for probabilistic forecasts. I do not think that is correct as these can also be developed using deterministic forecasts. P12L17: has no skill, and is equivalent to a random forecast P13F4: The map is very small, making it difficult to read. Consider increasing its size. P13L16: In the proximity of southern Africa PL13: Chockwé is on the main river and therefore does not represent a sub-basin. P13L20: given the large sample size of 724 observations. What are these observations? Please explain. Are these months, or days? P14L16: Here some of the indicators are discussed before they are introduced. Perhaps add references to the tables here. P16Fig7: What is SRI_NOW? Is this the standardised runoff? I guess so – please clarify. Also clarify what is meant by interactions of selected predictors (grey). P16: It is not so clear what the differences are between ERSST and OISST. Please explain (briefly). These also achieve very different results. P16: In the discussion it is mentioned that the selection of the indicators is unexpectedly low in some cases, which is due to the low correlation. However, this may also be the case for the superiority of Darwin SLP over ENSO. Please try and generalise such findings.

P18L1-3: The results in the figure shows that the importance may vary quite dramatically at the same location during the year. This is not really explained (except that it is very changeable). Is this seasonality? P18L33: At several stations P20L3: also exhibit a strong P20L7: However, our study suggests P21L31: at a lower level P22L7: The discussion on if the errors are small then there is skill seems somewhat trivial. But maybe there is something missing? P23L9: here it is suggested to explicitly consider

the human influence. I cannot agree more. However, I am not sure what is meant by: with the scope on the role of..please clarify P24Fig14: This figure is small and difficult to read. P25: The conclusions can be improved, primarily in writing style. The current style is very staccato and does not flow well. Try and make a bit more of an essay./strotyline. .
* * *

---

## Author Comment (AC1) · 24 May 2016

[10pt,a4paper,oneside]article [utf8]inputenc amsmath amsfonts amssymb

**Author's response to Anonymous Referee #1**

Mathias Seibert

May 24, 2016

Dear referee,

we would like to thank you very much indeed for your comments on our manuscript.

Reply to the original comments:

1. Page 7, line 3: "The standardised streamflow indices (SSI) are calculated for each station at the scale of 6 months. $SSI_6^May$ of May at that scale covers the desired main runoff period from December to May, henceforth named $SSI_{DJFMAM}$ (Figure 2)". When discussing the SSI, it is not clear if the SSI is a single value (averaged or summation?) for the months Dec-May or each month has its own SSI value. I would assume that there is only one SSI value for Dec-May, in that case Figure 2 is showing the Box-Plots of Monthly streamflow and not the SSI. I don't see the use of Figure 2 in this manuscript in relation to SSI.

   - Agreed. The figure does not directly help the reader to better understand the SSI. However, it was meant to help the reader to understand what we defined as the "desired main runoff period". While the complete removal of the figure could benefit the overall length of the paper we would like the
promote its use as a way to inform the reader about the region's seasonal regime, helping those not familiar with Southern Africa's climate. Therefore, we moved the figure reference to the description of the study area (section 2.1, page 4, line 16).

Regarding your critique that it is not clear whether "SSI is a single value...or each month has its own SSI value", we added the following two sentenses at the beginning of the section, hoping to clarify this: "In streamflow standardisation a time series is transformed to a normally distributed time series, which can be applied at different temporal scales. At the chosen scale, the respective period (for example January-February) is averaged annually and then standardised based on all annual values present in the time series."

2. Page 13, line 2: " Therefore, the RFOR predictor importance was modified for comparison ." How was it modified?

- To be very clear and avoid confusion for the reader, we have deleted the first two sentences of this paragraph, so that we start directly with collinearity and then explain, how predictor importance is calculated: "Collinearity of predictors can affect the importance estimation, since predictors might easily replace each other in the regression trees if they have a similar predictive strength. This can cause several effects. On the one hand, the importance per single predictor might be underestimated, if it is not located at an important position in all regression tree models. On the other hand, in presence of collinearity there would be multiple predictors with underestimated predictor importance. Therefore, the results of RFOR predictor importance are summarised for comparison with the MLM partial coefficient of determination. Closely related predictors are merged as relative group importance, calculated as ..." (page 13, lines 10-15)

3. Suggestion: As ANN is not bound by any linear assumptions (as opposed to

MLM), the use of the MLM predictors which were selected based on Pearson correlation (a linear technique) and relying on MLM stepwise predictor selection has limited the performance of ANN in this study. I suggest that in the future studies, the authors do not bound ANN to limited linear selection of inputs (predictors) and investigate a wider range of inputs using either a simple method of trial and error with ANN or more complicated methods such as mutual information or genetic algorithm to select ANN's inputs.

- Indeed, it is likely that the ANNs performance might have been reduced by the chosen predictor selection. In future studies we will prefer to keep MLM and ANN predictor selections completely independent.

The authors would like to express their appreciation for the received revisions and suggestions. Thank you very much.

[Figure]

**Supplement:**

[revised manuscript text omitted]

---

## Author Comment (AC2) · 24 May 2016

Dear referee,

we would like to thank you very much indeed for your comments on our manuscript. Please check the attached pdfs

Reply to the main comments:

1. I would also like to ask the authors to carefully revise how they refer to the various indices, which is somewhat confusing at times. What confused me is that there are standard indices in both Atlantic and Indian Ocean, but also customized indices. The latter are then referred to as Atlantic and Indian Ocean only. I would propose that the authors revise this and always precede the customised indices with the word "Custom" or something like that. That would help clarify somewhat to my mind.

- We went through the manuscript checked the mentioning of Ocean regions for correctness. Here's the result:
- We added a clear connection in descriptions of figures 8 and 9 so that the "Atlantic" and "Indian Ocean" predictor groups are customised indexes.

2. Overall the figures in the manuscript could be made a little larger to enhance visibil- ity/interpretation. There are some very small figures, and at times the figures are not easy to read (e.g Figure 10 could be improved by plotting the thick black line differently).

- Figure 4: Has been increased to full page width improve readability.
- Figure 10: Unfortunately we were not fully able to understand the reviewers request to plot the black line "differently" in Figure 10. We understand that the overlay of several lines makes it hard to distinguish the lines. Yet, after all, we trust the reader is able to comprehend, that an invisible line color means it is the same as the line above.

3. My main comment on the paper is the influence of the dams within the catch- ment is not well explored. In some places the authors elude to the presence of dams, and also include details as to their total volume compared to the average annual volume that enters the dam. This shows that for some of the stations the anthropogenic influence is substantial. In many cases there is more storage than there is annual volume, such as is the case for Nauwpoort. And yet this is one of the two stations that are reported to have the highest skill (together with Hartbeeshoek, which has no upstream, dams). This is surprising. This is also

linked to one of the findings of the authors that the predictability of the smaller catchments is better than for the larger catchments. This is an interesting conclusion because it is somewhat counterintuitive, because as the authors note, the lower skill in large catchments may be due to the anthropogenic influences. However, these catchments are really very small. This would mean that the skill found cannot be due to the persistence of the catchment initial conditions.

- The comment on the influence of dams is well taken. Dams are abundant in the basin and we have knowledge of 55 dams built from 1929 until 2012 with capacity information, but the list is unlikely to be complete. From 1929 to 1976 the total dam capacity in the Limpopo basin increased by about 35 $Mm^3a^{-1}$, then in 1976 the Massingir dam was built adding 2800 Mm3. Thereafter, the construction rate slightly increased to $39 Mm^3a^{-1}$, most likely also as a consequence of the catastrophic drought events in the 80's and 90's. The total dam capacity today is about 6500 $Mm^3$. We suppose that many more unregulated and small dams exist. Often, dams serve as reservoir for irrigation and household use. In addition, streamflow abstraction for irrigation is a common water source for farmers, beside groundwater. However, information on irrigation amounts is rare. Further human intervention are water transfers, for example in Botswana: Intrabasin from Francistown to Gaborone, and interbasin from the Okawango to the Limpopo.

  Dams, abstractions and transfers create a complex picture of anthrogenic interference which is very complicated to disentangle - if not impossible - even with a hydrological model, since data availability is low. Therefore, without reliable data to support a proper analysis, we could only speculate why some stations show better results. To stress the importance of human interventions in relation to seasonal forecasts, we extended the discussion in the last paragraph of the discussion section on page 23 (from line 32).

4. The last overall comment I have is on the selection of the customised indices.

The authors note that these were selected over a large area. However, I can imagine that there is a trade-off between the large area and the ability to find significant differences/anomalies. I would expect that as the area gets larger, the detection of anomalies gets smaller. Perhaps the authors could comment on this.

- There definitely is a trade-off between capturing location and strength of an important ocean region. SST anomalies are not bound to a specific location. Every event has its own genesis resulting in a different spatial pattern. Both methods, correlation and composite analysis are used to find regions that are repeatedly covered by the different past events. These analyses was performed for different time windows and lead times. Yes, it would be possible to create an index for every exact location (polygon) resulting from the analysis. However, this would have resulted in way to many potential predictors, which would have required a reduction in dimensionality, for example with a principal component analysis. Principal components are practical, yet more complicated to grasp and interpret in the end. Therefore we argue that the proposed method is well justified, providing a compromise between preciseness of predictor locations and regions on the one hand, and interpretability of the results.

Reply to the specific comments:

| | |
|---|---|
| P1L4: assessed using statistical | corrected |
| P1L9: as a proxy | corrected |
| P1L15: warning, the models | corrected |
| P2L5: which have severe | unchanged, this is referring to the past events in the 80's and 90's |
| P2L8: which may even | corrected |
| P2L9: regarded as being highly affected | corrected |
| P2L11: to studies that found | corrected |
| P2L10-12: There is some discussion on the climate. I am not sure these comments are entirely relevant to this manuscript. | The intention was to give a background on climate change in the Limpopo region, event though this study is not about climate change. However, seasonal forecasting is a potential adaptation strategy for drought prone regions, such as Southern Africa. Shortened the discussion by one sentense. |
| P3L1: Atmospheric circulation processes have . . . | corrected |
| P3L6: it extends from the ocean | corrected |
| P3L13: by the chaotic | corrected |
| P3L26: These are particularly | corrected |
| P3L33: The skill of the forecasting | corrected |

P3L33: The authors refer to the DJFMAM forecast. That is clear that this spans the wet season. But is this a single value, or is there a forecast for each month. Perhaps I missed it, but it may be good to clarify in the text what a forecast actually contains in terms of parameters and time steps.

In the publication by Trambauer et al, that we are referring to, they have several forecasts. The one we are referring to is the lead time of five months for May, which is only one value per year. The sentence was changed to: "The skill of the forecasting system for total streamflow between December and May (DJFMAM) exceeded climatological forecasts (climatology) with "moderate skill for all lead times" up to 5 months (forecast in December)"

P5L3: There is some discussion on extracting the catchment areas. Why are these relevant other than to be included in the table describing the catchments.

The sentence names the data source, that was used to derive the catchment area and other GIS tasks. It has no greater relevance to the study.

P5L12: event anomalies

corrected

P6 Table 1: It may be useful to include the year in which the dam was built, or at least the main dam building period in the Limpopo. This can help interpret possible issues of stationarity in the time series.

Due to the high number of dams, there rarely is a single date for dam construction. Thus, this information is hard to reduce for a single column. Dam construction and management definitely causes instationarity in the time series.

P7L20: with df = N-2 degrees of freedom

corrected

P7L23: The region outlines

corrected

P7L23: generously, so as to

corrected

P8Table3: It is not so clear what the aggregation period of the streamflow indices is. Are these for one month? Or rather is the SSIDJMAM the ag- gregated streamflow index across the whole wet season. This should all be clarified a bit better.

$SSI_{DJFMAM}$ is a single value per year. However, The table is meant to describe the lead time definition and is not a good place for the SSI description, which was moved to the beginning of section 2.3 and the 2nd paragraph of section 2.5 (model setup).

P9L2: linear regression is applied to estimate the values of parameters Bo to Bp.

corrected

P9L10: until the addition or removal does not lead to an increase in model quality.

corrected

P10L20: The hidunitj variable is somewhat long and should be avoided. Perhaps introduce something simpler, such as H, and explain it well.

corrected

P11: I was not so clear how the forecast skill of the ANN is expressed, and if that is commensurate with how it is expressed for the criteria used to establish the MLM parameters. Please ensure that these are well defined, and that that if there are differences explain why the calibrated models may then be compared.

All methods undergo leave-one-out cross validation, the result of which is used to express the forecast skill. A respective paragraph was added at the end of section 2.6 on page 12, lines 29 to 31

P11L22: The trees are trained

This is jargon applied to trees and mashine learning. However, I understand the confusion very well and changed it.

P11L26: I am not sure what is meant by the final node size.

It is a technical term. The dataset is split into branches to reduce variation within the groups aka nodes. These groups must have more than 5 samples. It is not possible to pick a group of one to accommodate an outlier, for example. The description of Randomforest was improved to accomodate for that.

P11: Overall the description of the Random Forest Trees is difficult to follow for those not familiar. What are the 500 regression trees? What is the minimum final node size? I think the majority of the readers of HESS will not be familiar with this technique. ANN is more familiar I think. The authors use quiet a lot of jargon such as "bagging" etc. I would be very helpful if they provide a simple explanation of this technique and how a forecast is actually derived.

We improved the explanation of Randomforest, particularly for readers, unacquainted with the method. However, details must we left for specific literature and papers such as Breiman (2001).

P12L1: model over fitting (I changed this but please check the context)

No, here, overfitting is not correct in this place. Overfitting is not a desirable characteristic for models, but model data fit to the measurement data is.

P1211: It is suggested that 2x2 contingency tables can be used only for probabilistic forecasts. I do not think that is correct as these can also be developed using deterministic forecasts.

I was unable to find that statement in line 11. We merely describe how contingency tables were constructed for the ROC analysis for probabilistic forecasts, which changed to be more clear. No doubt, there are methods for deterministic forecasts, too.

P12L17: has no skill, and is equivalent to a random forecast

corrected

P13F4: The map is very small, making it difficult to read. Consider increasing its size.

corrected

P13L16: In the proximity of southern Africa

corrected

PL13: Chockwé is on the main river and therefore does not represent a sub-basin.

corrected

P13L20: given the large sample size of 724 observations. What are these observations? Please explain. Are these months, or days?

These are months. Corrected.

P14L16: Here some of the indicators are discussed before they are introduced. Perhaps add references to the tables here.

The sentense was rephrased to introduce the regions and a reference was added: "Nevertheless, the currents themselves are represented by customised predictors based on other ocean regions in the Indian Ocean (predictor named "Agu") and the southern Atlantic (predictors named "SWAtl", "SEAtl", "BC" in figure 6)."

P16Fig7: What is SRI_NOW? Is this the standardised runoff? I guess so - please clarify. Also clarify what is meant by interactions of selected predictors (grey).

Yes, SRI_NOW is the current streamflow index. I added a reference in the figure description, also for the MLM interactions.

P16: It is not so clear what the differences are between ERSST and OISST. Please explain (briefly). These also achieve very different results.

These are both SST datasets. The OISST data set includes additional observations, such as satellite imagery and others, instead of buoy and ship observations only. The data quality is supposed to be better, with the major disadvantage of a shorter time span. ERSST is selected more often. We extended the description a little bit, but do not consider it worth a more detailed discussion with regard to the general question.

P16: In the discussion it is mentioned that the selection of the indicators is unexpectedly low in some cases, which is due to the low correlation. However, this may also be the case for the superiority of Darwin SLP over ENSO. Please try and generalise such findings.

We moved this discussion to a separate subsequent paragraph to give it more emphasis and rephrased it. However, this study is not designed to generally and finally distinguish the influence of DARWIN SLP vs. ENSO on the southern African region and - the results from this study cannot really negate Manatsa et al. (2007). However, our result is definitely not creating further evidence for the claim by Manatsa et al. (2007).

P18L1-3: The results in the figure shows that the importance may vary quite dramatically at the same location during the year. This is not really explained (except that it is very changeable). Is this seasonality?

It is the part of the result that also gave us some headaches. Attribution is tricky. Some of those changes might be seasonal changes. However, much of it must also be considered random. One has to keep in mind: Most of these models only achieve a low total $R^2 < 0.3$. If a predictor reaches 0.1 in relative partitioned $R^2$, i.e. and the total explained variance is only 30 %, then that particular predictor explains only about 3%. Thus, one should try to find the overarching pattern and not interpret specific contributions at certain lead times. One might easily overinterpret the numbers. Therefore, we did not go into more detail, here. (respective discussion added on page 19, line 1)

| | |
|---|---|
| P18L33: At several stations | corrected |
| P20L3: also exhibit a strong | corrected |
| P20L7: However, our study suggests | corrected |
| P21L31: at a lower level | corrected |
| P22L7: The discussion on if the errors are small then there is skill seems somewhat trivial. But maybe there is something missing? | Trivial, yet enlightening. The error does not seem to be constant with all observations. For stations with large errors a few events have a high influence on the skill outcome. From this observation one can conclude that the time period of 30 years of observation is not long enough to derive robust forecast models. |
| P23L9: here it is suggested to explicitly consider the human influence. I cannot agree more. However, I am not sure what is meant by: with the scope on the role of..please clarify | What was meant is: "focussing on the role of...", changed accordingly. |
| P24Fig14: This figure is small and difficult to read. | corrected |
| P25: The conclusions can be improved, primarily in writing style. The current style is very staccato and does not flow well. Try and make a bit more of an essay./stroryline. | We deliberately chose a short and straight style for the conclusions and would very much like to keep it that way. We hope to inform the reader quickly about the major lessons to learn from this work, but indeed, it ended up a bit staccato. We gave it a few minor touches to improve the flow, but would very much like to keep the general structure. |
The authors would like to express their appreciation for the received revisions. Thank you very much.

PS: Please check the changes made in the attached manuscript update.

---

## Author Response (AR2)

**Author's response to Editor review**

Mathias Seibert

October 3, 2016

Dear editor,

we would like to thank you very much for your comments on our manuscript. Please see below our reply to the two comments:

1. On the discussion of the influence of major dams, as well as the finding that the forecasting skill of the smaller catchments is better than the larger I find their response weak. A suggestion has also been added on the reliability of the rating curves due to changes in the cross section. While I agree this is the case, I think it is somewhat suggestive. The rating curve at for example Chokwé is indeed susceptible to change due to changing cross section, but I think this applies primarily for higher flow conditions. It is not clear on what data/information they base that suggestion.

   - We understand the question is referring to the sentence in lines 14 and 15 on page 25 which is part of a paragraph discussion the sources of forecast uncertainty in this study. With the sentence we like to make the reader aware of the uncertainty in streamflow measurements in general. We strongly agree with the reviewer, that changes of river cross sections can have a strong effect on high flows. In some places also low flows can also be very uncertain. Unfortunately, due to the unavailability of data, we cannot give any information regarding to the changes of river cross sections in the stations regarded in this study. In order to clarify, that here, we are not referring to individual dams but discussing the sources of uncertainty we would like to rephrase the sentence "Furthermore, there is always a high uncertainty in streamflow measurement, which rely on repeatedly updated river cross sections" to "Furthermore, streamflow observation in general is subject to high uncertainty.".

2. Some of the changes made require revision in terms of language. Some of the new sentences are confusing. In one case on page 16 the authors added a discussion on ERSST and OISST. In subsequent sentences the preferred choice is indicated, but that choice does differ per sentence. That is somewhat confusing.

   - The sentence was added to explain the difference of ERSST and OISST in condensed way and the respective result of the predictor selection. In order to better convey the defference of the data sets on the one hand and the results of the predictor selection on the other hand, the sentences are changed to: "ERSST would be the recommended data set for time series modelling due to its greater length of

record. On the other hand, the OISST data set is a shorter data set with higher quality achieved by the inclusion of new and improved types of SST observations such as satellite imagery. However, the results show that the ERSST ENSO indices are selected more often than the OISST ENSO indices."

The authors would like to express their appreciation for the received comments. Thank you very much.